# Regional pollen-based Holocene temperature and precipitation patterns depart from the Northern Hemisphere mean trends

Ulrike Herzschuh[1,2,3], Thomas Böhmer[1], Manuel Chevalier[4,5], Raphaël Hébert[1], Anne Dallmeyer[6], Chenzhi Li[1,2], Xianyong Cao[1,7], Odile Peyron[8], Larisa Nazarova[1,9], Elena Y. Novenko[10,11], Jungjae Park[12,13], Natalia A. Rudaya[14,15], Frank Schlütz[16,17], Lyudmila S. Shumilovskikh[17], Pavel E. Tarasov[18], Yongbo Wang[19], Ruilin Wen[20,21], Qinghai Xu[22], Zhuo Zheng[23,24]

[1] Polar Terrestrial Environmental Systems, Alfred Wegener Institute Helmholtz Centre for Polar and Marine Research, Telegrafenberg A45, 14473 Potsdam, Germany

[2] Institute of Environmental Science and Geography, University of Potsdam, Karl-Liebknecht-Str. 24–25, 14476 Potsdam, Germany

[3] Institute of Biochemistry and Biology, University of Potsdam, Karl-Liebknecht-Str. 24–25, 14476 Potsdam, Germany

[4] Institute of Geosciences, Sect. Meteorology, Rheinische Friedrich-Wilhelms-Universität Bonn, Auf dem Hügel 20, 53121 Bonn, Germany

[5] Institute of Earth Surface Dynamics IDYST, Faculté des Géosciences et l'Environnement, University of Lausanne, Bâtiment Géopolis, 1015 Lausanne, Switzerland

[6] Max Planck Institute for Meteorology, Bundesstraße 53, 20146 Hamburg, Germany

[7] Alpine Paleoecology and Human Adaptation Group (ALPHA), State Key Laboratory of Tibetan Plateau Earth System, Resources and Environment (TPESRE), Institute of Tibetan Plateau Research, Chinese Academy of Sciences, 100101 Beijing, China

[8] Institut des Sciences de l'Evolution de Montpellier, Université de Montpellier, CNRS UMR 5554, Montpellier, France

[9] Institute of Geology and Petroleum Technologies, Kazan Federal University, Kremlyovskaya Street 18, 420008 Kazan, Russia

[10] Faculty of Geography, Lomonosov Moscow State University, Leniskie Gory 1, 119991 Moscow, Russia

[11] Department of Quaternary Paleogeography, Institute of Geography Russian Academy of Science, Staromonrtny Lane 29, 119017 Moscow, Russia

[12] Department of Geography, Seoul National University, 1 Gwanak-ro, Gwanak-gu, Seoul 08826, Republic of Korea

[13] Institute for Korean Regional Studies, Seoul National University, 1 Gwanak-ro, Gwanak-gu, Seoul
08826, Republic of Korea
[14] PaleoData Lab, Institute of Archaeology and Ethnography, Siberian Branch, Russian Academy of
Sciences, Pr. Akademika 36 Lavrentieva 17, 630090 Novosibirsk, Russia
[15] Biological Institute, Tomsk State University, Pr. Lenina, 26, 634050 Tomsk, Russia
[16] Institute for Pre- and Protohistoric Archaeology, University of Kiel, Germany, Johanna-Mestorf-Straße
2–6, 24118 Kiel, Germany
[17] Department of Palynology and Climate Dynamics, Georg-August-University of Göttingen, Wilhelm
Weber Str. 2a, 37073 Göttingen, Germany
[18] Institute of Geological Sciences, Palaeontology Section, Freie Universität Berlin, Malteserstraße 74–
100, Building D, 12249 Berlin, Germany
[19] College of Resource Environment and Tourism, Capital Normal University, 105 West Third Ring Road
North, 100048 Beijing, China
[20] Key Laboratory of Cenozoic Geology and Environment, Institute of Geology and Geophysics, Chinese
Academy of Sciences, 19 Beitucheng West Road, Chaoyang District, 100029 Beijing, China
[21] Center for Excellence in Life and Paleoenvironment, Chinese Academy of Sciences, 100044 Beijing,
China
[22] School of Geographic Sciences, Hebei Normal University, 050024 Shijiazhuang, China
[23] Guangdong Key Lab of Geodynamics and Geohazards, School of Earth Sciences and Engineering,
Sun Yat-sen University, 519082 Zhuhai, China
[24] Southern Marine Science and Engineering Guangdong Laboratory (Zhuhai), 519082 Zhuhai, China
*Correspondence to:* Ulrike Herzschuh (Ulrike.Herzschuh@awi.de)

**Abstract.** A mismatch between model- and proxy-based Holocene climate change, known as the
´Holocene conundrum´, may partially originate from the poor spatial coverage of climate reconstructions
in, for example, Asia, limiting the number of grid cells for model-data comparisons. Here we investigate
hemispheric, latitudinal, and regional mean time-series as well as time-slice anomaly maps of pollen-
based reconstructions of mean annual temperature, mean July temperature, and annual precipitation
from 1908 records in the Northern Hemisphere extratropics. Temperature trends show strong latitudinal
patterns and differ between (sub-)continents. While the circum-Atlantic regions in Europe and Eastern
North America show a pronounced Mid-Holocene temperature maximum, Western North America
shows only weak changes and Asia mostly shows a continuous Holocene temperature increase.
Likewise, precipitation trends show certain regional peculiarities such as the pronounced Mid-Holocene
precipitation maximum between 40 and 50°N in Asia and Holocene increasing trends in Europe and
Western North America, which can all be linked with Holocene changes of the regional circulation pattern
responding to temperature change. Given a background of strong regional heterogeneity, we conclude
that the calculation of global or hemispheric means, which initiated the ´Holocene conundrum´ debate,
should focus more on understanding the spatio-temporal patterns and their regional drivers.

**1 Introduction**
Previous comparisons of proxy-based reconstructions and simulations of global Holocene climate
change have yielded major mismatches, a discrepancy termed the ´Holocene conundrum´ (Liu et al.,
2014c; Kaufman and Broadman, 2023). While simulations indicate an increase in Holocene temperature
(Liu et al., 2014c), proxy data syntheses rather support a Mid-Holocene temperature maximum (Marcott
et al., 2013; Kaufman et al., 2020b). Recently, several explanations for this finding were proposed, most
of which assign the mismatch to biases in the proxy data with respect to location or seasonality (Marsicek
et al., 2018; Bader et al., 2020; Bova et al., 2021; Osman et al., 2021).
Previous temperature reconstructions from continental areas are mainly available from the circum-North
Atlantic region, and are potentially unrepresentative of the whole Northern Hemisphere temperature
change, as the region was strongly impacted by the vanishing Laurentide ice-sheet (Rolandone et al.,
2003; Chouinard and Mareschal, 2009). Synthesis studies hitherto included rather few records from the
large non-glaciated Asian continent (Andreev et al., 2004; Leipe et al., 2015; Melles et al., 2012;
Nakagawa et al., 2002; Stebich et al., 2015; Tarasov et al., 2009 and 2013). The inclusion of recently
compiled Holocene pollen records (Cao et al., 2019; Herzschuh et al., 2019) and high-quality modern
pollen datasets (Tarasov et al., 2011; Cao et al., 2014; Davis et al., 2020; Dugerdil et al., 2021) from
Asia now allows for higher quality quantitative reconstructions.
While temperature patterns have often been studied, hemispheric syntheses of quantitative precipitation
change during the Holocene are not yet available. A recent study of qualitative moisture proxy data
suggests an overall warm and dry Mid-Holocene in the Northern Hemisphere mid-latitudes, related to
the weakened latitudinal temperature gradient (Routson et al., 2019). This trend contrasts with the idea
of positive hydrological sensitivity, that is, warm climates are wet at a global scale (Trenberth, 2011),
which was confirmed from proxy and model studies from monsoonal areas in lower latitudes (Kutzbach,
1981; Wang et al., 2017). However, the study of Routson et al. (2019) only included a few records from
the subtropical monsoonal Asia that is known for complex Holocene moisture patterns (Herzschuh, 2004;
Chen et al., 2019; Herzschuh et al., 2019). These and further synthesis studies (Wang et al., 2010; Chen
et al., 2015; Wang et al., 2020) also gave a plethora of alternative explanations to characterize these
patterns, including interactions between the monsoon and westerlies circulation and evaporation effects.
Pollen spectra are a well-established paleoclimate proxy and quantitative estimates of past climatic
change are mainly derived by applying (transfer functions of) modern pollen-climate calibration sets to
fossil pollen records (Birks et al., 2010; Chevalier et al., 2020). Accordingly, pollen-based
reconstructions constitute a substantial part of multi-proxy syntheses (e.g., Kaufman et al., 2020b), albeit
derived from different calibration sets and methods, which makes a consistent assessment of inherent

reconstruction biases difficult. Pollen data are one of the few land-derived proxies available that can theoretically contain independent information on both temperature and precipitation in the same record (Chevalier et al., 2020; Mauri et al., 2015). Consistent pollen-based reconstructions can thus contribute to better characterizing past temperature and precipitation changes across large landmasses and how these changes co-vary over time (Davis et al., 2003).

Here, we analyze spatio-temporal patterns of pollen-based reconstructions of mean annual temperature ($T_{ann}$), mean July temperature ($T_{July}$), and mean annual precipitation ($P_{ann}$) from 1908 sites from the Northern Hemisphere extratropics that were generated using harmonized methods and calibration datasets (LegacyClimate 1.0, Herzschuh et al., 2022a) and have revised chronologies (Li et al., 2022). We address the following questions: (1) What are the continental, latitudinal, and regional patterns of Holocene temperature change in the Northern Hemisphere extratropics and how do our new reconstructions align with the global averaged trends of a previous global temperature synthesis? (2) What are the continental, latitudinal, and regional patterns of Holocene precipitation change and how do these changes co-vary with temperature trends?

## 2 Methods

This study analyzes pollen-based reconstructions provided in the LegacyClimate 1.0 dataset (Herzschuh et al., 2023). It contains pollen-based reconstructions of $T_{July}$, $T_{ann}$, and $P_{ann}$ of 2593 records along with transfer function metadata and estimates of reconstruction errors and is accompanied by a manuscript analyzing reconstruction biases and presenting reliability tests (Herzschuh et al., 2022a). The fossil pollen records, representing the LegacyPollen 1.0 dataset, were derived from multiple natural archives, most commonly continuous lacustrine and peat accumulations (Herzschuh et al., 2022b), and originate from the Neotoma Paleoecology Database (´Neotoma´ hereafter; last access: April 2021; Williams et al., 2018), a dataset from Eastern and Central Asia (Cao et al., 2013; Herzschuh et al., 2019), a dataset from Northern Asia (Cao et al., 2019), and a few additional records to fill up some spatial data gaps in Siberia.

The chronologies of LegacyPollen 1.0 are based on revised ´Bacon´ (Blaauw and Christen, 2011) age-depth models with calibrated ages at each depth provided by Li et al. (2022). Taxa are harmonized to genus level for woody and major herbaceous taxa and to family level for other herbaceous taxa. Along with LegacyClimate 1.0, a taxonomically harmonized modern pollen dataset is provided (a total of 15379 samples; Herzschuh et al., 2022a) which includes datasets from Europe (EMPD2, Davis et al., 2020), Asia (Tarasov et al., 2011; Herzschuh et al., 2019; Dugerdil et al., 2021), and North America (from Neotoma; Whitmore et al., 2005). LegacyClimate 1.0 also provides the climate data for the sites of the modern pollen samples that were derived from WorldClim 2 (Fick and Hijmans, 2017).

LegacyClimate 1.0 provides reconstructions based on different methodologies including two versions of WA-PLS (weighted averaging partial least squares regression, a transfer function-based approach) and MAT (modern analogue technique). For each fossil site, we calculated the geographic distance between each modern sampling site and each fossil location and selected a unique calibration set from modern

sites within a 2000 km radius (Cao et al., 2014), as it was shown to be a good trade-off between analog
quality and quantity (Cao et al., 2017). For WA-PLS, the used component, typically first or second, was
identified using model statistics as derived from leave-one-out cross-validation based on the criterion
that an additional component be used only if it improves the root mean squared error (RMSE) by at least
5% (ter Braak and Juggins, 1993). A WA-PLS_tailored reconstruction is also provided in the
LegacyClimate 1.0 dataset (Herzschuh et al., 2022a), which addresses the problem that co-variation in
modern temperature and precipitation data can be transferred into the reconstruction. To reduce the
influence of one climate variable on the target variable, the modern range of the non-target variable is
reduced by tailoring the modern pollen dataset to a selection of sites with little covariance between the
two variables. For example, to reconstruct $T_{July}$ we identified the $P_{ann}$ range reconstructed by WA-PLS
and extended it by 25% at both ends. For the selection of sites in the modern training dataset, we then
restricted modern $P_{ann}$ to that range accordingly. As such, we keep all information for reconstruction
from those modern pollen spectra that cover a wide temperature range but downweight the information
from pollen spectra covering a wide precipitation range. However, initial assessments did not show any
major differences compared to using the standard WA-PLS-derived reconstruction. Therefore, we do
not make use of this dataset for this study so as to be consistent with previous studies. For comparison,
we provide a plot with hemispheric, continental, and latitudinal mean curves for $T_{July}$, $T_{ann}$, and $P_{ann}$
reconstructed by WA-PLS_tailored in the supplement. The MAT reconstructions were derived from the
seven best analogs that we identified based on the dissimilarity measures between the fossil samples
and the modern pollen assemblages using the squared-chord distance metric (Simpson, 2012). MAT
reconstructions were highly correlated with those obtained by WA-PLS (Herzschuh et al., 2022a). Here,
we opted for the widely used WA-PLS, as it is less sensitive to the size and environmental gradient
length of the modern pollen dataset and is thus less affected by spatial autocorrelation effects and can
better handle poor analog situations (ter Braak and Juggins, 1993; Telford and Birks, 2011; Cao et al.,
2014; Chevalier et al., 2020). Statistical significance tests sensu Telford & Birks (2011) were performed
for each site for WA-PLS, WA-PLS_tailored and MAT and assessed in Herzschuh et al. (2022a).
Of the 2593 records available in LegacyClimate 1.0, 1908 records with at least 5 samples that cover at
least 4000 years of the Holocene and have a mean temporal resolution of 1000 years or less were
included in the time-slice comparisons based on this criterion (Fig. 1). The construction of time-series
to estimate the means of climate variables was further restricted to 957 records that cover the full period
of 11 to 1 ka.

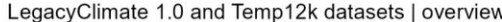

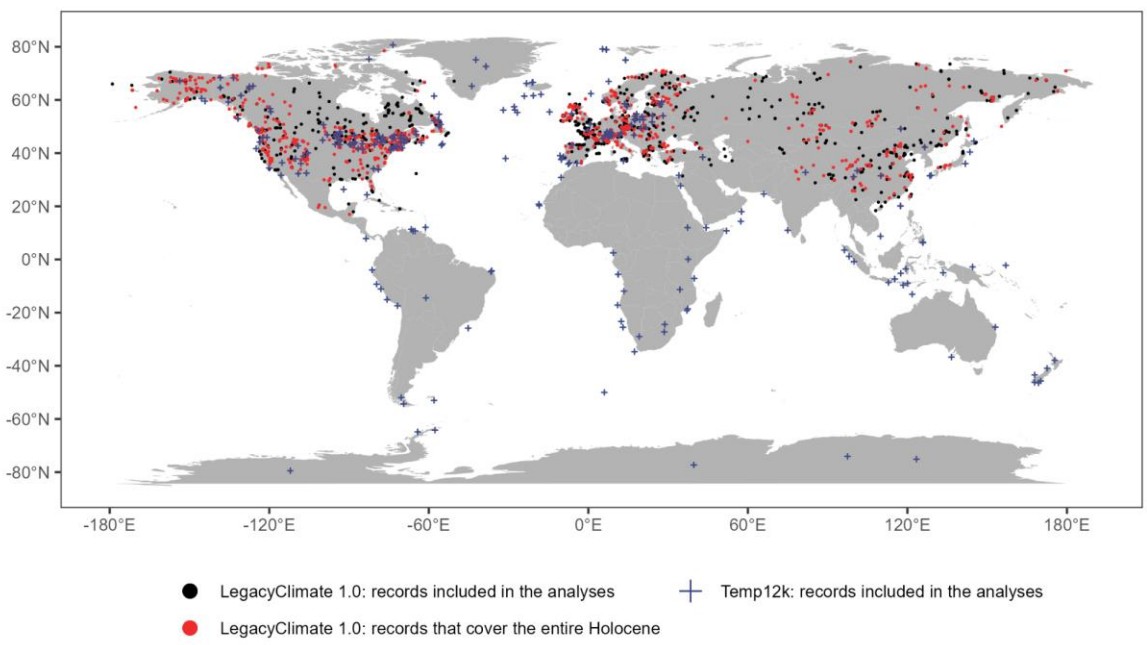

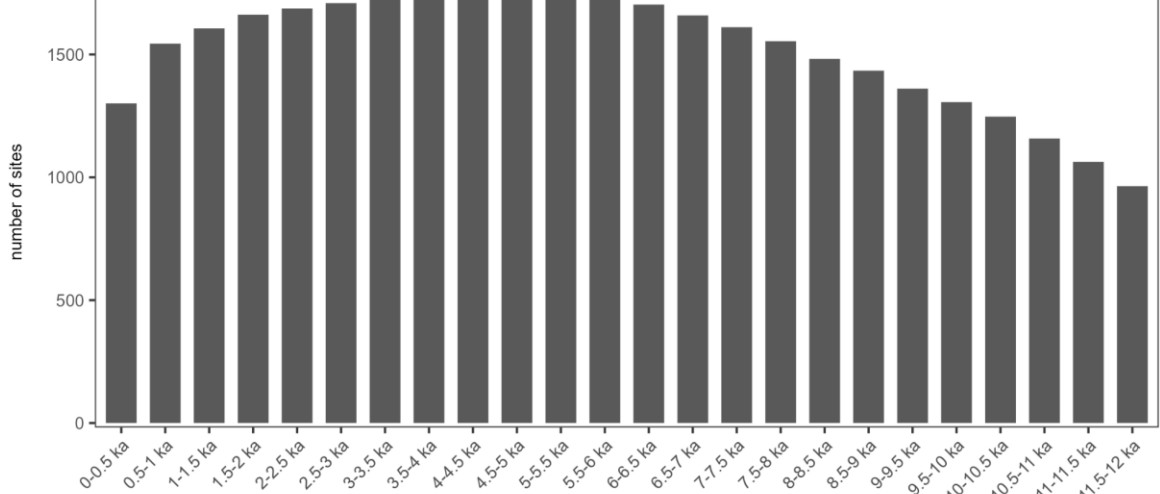



**Figure 1. (top) Spatial coverage of the LegacyClimate 1.0 (dots) and Temp12k (Kaufman et al. 2020b, crosses) datasets used in this analysis.** The map shows sites that cover the entire Holocene (i.e., 11-1 ka) as red symbols and those that cover parts of the Holocene but at least 4000 years in the period between 12 and 0 ka as black symbols. (bottom) Temporal coverage of the LegacyClimate 1.0 dataset.

181

The mean root mean squared error of prediction (RMSEP; WA-PLS) from all 957 sites included in the time-series analyses is 2.4±0.7°C (one standard deviation) for $T_{July}$, 2.6±0.5°C for $T_{ann}$, and 244±74 mm for $P_{ann}$. They show a spatial pattern in that the RMSEPs are higher in areas with steep climate gradients (e.g. Central Asia and along the western coast of North America; see Fig. 5 in Herzschuh et al., 2022a).

As it has already been shown in previous comparisons, WA-PLS can have higher RMSEPs than MAT
but these do not necessarily reflect a less reliable reconstruction but methodological differences. MAT
is known to be more sensitive to spatial autocorrelation, which causes the model performance to be
over-optimistic compared to WA-PLS (Cao et al., 2014). Besides, trends and the relative changes, as
interpreted in this study, are less sensitive to methodological biases than absolute values.
Derived time-series of $T_{July}$, $T_{ann}$, and $P_{ann}$ were smoothed over a 500-yr time-scale and resampled at a
100 yr-resolution using the *corit* package in R (version 0.0.0.9000, Reschke et al., 2019). Because the
original time-series are unevenly spaced, we used this package as it is designed to resample irregularly
sampled time-series to an equidistant spacing (Reschke et al., 2019). The smoothing length of 500
years reflects the typical resolution of the original pollen records. These derived time-series were
sampled at selected time-slices and converted into a regular 2° x 2° raster grid (by taking the mean of
all records located within the grid cell) using the *raster* package in R (version 3.5-11, R Core Team,
2020; Hijmans et al., 2021).
To calculate zonal, (sub-)continental (i.e., Asia (>43°E), Europe (<43°E), Eastern North America
(<104°W; Williams et al., 2000) and Western North America), and hemispheric means we selected all
957 smoothed and resampled time-series of $T_{July}$, $T_{ann}$, and $P_{ann}$ that cover the full period between 11
and 1 ka and calculated climate anomalies for all three climate variables. Rather than using the
anomalies for $P_{ann}$ we calculated the precipitation change as % relative to the 1 ka reference period (Fig.
3) or relative to the younger time-slice (Fig. 4). The estimate at 1 ka was used as a reference to calculate
the anomalies, as many records either poorly or do not cover the last 0.5 ka. Weights proportional to the
inverse number of time-series per cell in the grid were used to calculate the weighted mean and standard
deviation (using the wtd.mean and wtd.var functions from the Hmisc R-package, version 5.0-1, Harrell
& Dupont, 2023). The weighted standard error was calculated by dividing the weighted standard
deviation estimates by the square root of the number of grid cells with at least 1 record. In total, 436 grid
cells between 17°N and 79°N are covered by one or more time-series (Fig. 2).
The zonal mean over 10° bands of (sub-)continents (e.g. for 30-40°N of Europe) were calculated and
also used to calculate the mean time-series of the (sub-)continents, with weights proportional to the
terrestrial area in a zonal band based on the WGS84 EASE-Grid 2.0 global projection (Brodzik et al.,
2012). Likewise, the area-weighting was applied to derive the continental means and hemispheric-wide
(zonal) means. We compare the linear trends of all zonal means with each other for each continent, as
well as the linear trends of the continental weighted means, taking into account the standard error of
each average. We take a Monte-Carlo approach to generate ensembles of trend estimates after adding
random errors and use a standard t-test to assess, pairwise, whether the means of the ensembles are
significantly different.

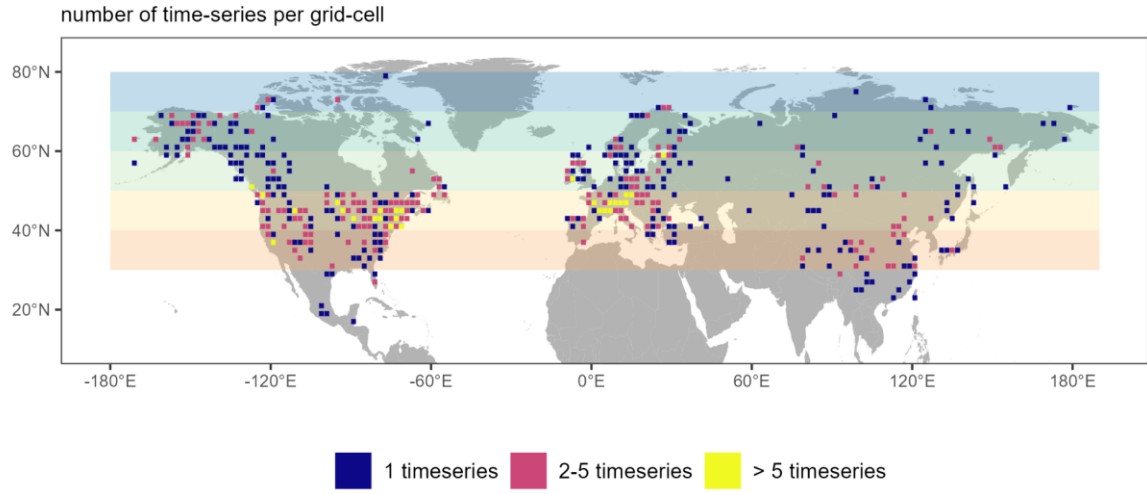

number of time-series per grid-cell

■ 1 timeseries    ■ 2-5 timeseries    ■ > 5 timeseries

**Figure 2. Number of time-series per grid cell.** The map shows the number of time-series that are
merged into one grid cell. Colored rectangles (as used for the zonal mean curves in Fig. 3) indicate the
latitudinal band a respective grid cell belongs to.

Furthermore, we extracted 325 records that cover the full Holocene period in the Temp12k dataset
(version 1-1-0; https://lipdverse.org/project/temp12k, last access February 2023; Kaufman et al., 2020b)
applying the same restrictions as with the LegacyClimate 1.0 dataset (i.e., at least 5 samples, a mean
temporal resolution of 1000 years or less). Instead of 11.0 ka we here used a cut-off of 10.5 ka as many
records in this dataset start shortly after 11.0 ka). For 43 sites, more than one temperature time-series
were stored in the Temp12k dataset. In these cases, we selected that time-series with the least amount
of missing temperature values in the period between 10.5 and 1 ka, leaving 272 records that were used
to construct the mean temperature anomaly time-series similar to the approach described for the
LegacyClimate 1.0 dataset. We excluded all pollen-based reconstructions from the Temp12k dataset
between 30°N and 80°N (n=117) to avoid duplications with the LegacyClimate 1.0 dataset when
integrating both datasets into a joint hemispheric and global mean temperature stack curve.

**3 Results**
**3.1 Spatio-temporal pattern of temperature reconstructions**
The temporal patterns of temperature records covering the entire Holocene (i.e., 11-1 ka) show strong
differences between continents (Fig. 3). Europe shows a pronounced Mid-Holocene temperature
maximum of +1.3±0.4°C for $T_{July}$ at 5.7 ka while the $T_{ann}$ maximum is less pronounced (+0.9±0.4°C at
5.8 ka). The Mid-Holocene $T_{July}$ was weaker and occurred earlier in Eastern North America (+0.5±0.2°C
at 7.0 ka) while $T_{ann}$ warming was +0.7±0.3°C at the same time period (7.0 ka). Asia ($T_{July}$) and Western
North America ($T_{ann}$) show almost no maximum but only some variations around a continuously
increasing Holocene trend, with a higher increase rate before 6 ka than after 6 ka.
Aside from these differences among (sub-)continents, certain regional differences exist. Early Holocene
cold climate anomalies were most pronounced in latitudes between 45°N and 65°N, particularly in
Northern Europe, Northeastern Asia, and Alaska (Fig. 4) with above 2.5°C deviation to Holocene $T_{ann}$
maximum values in most records. The most pronounced $T_{ann}$ maximum (more than 1.5°C warmer than
the Late Holocene) can be found in Europe north of 60°N and Eastern North America between 60°N
and 70°N, forming a circum-North Atlantic pattern (Fig. 5). Records from Eastern Europe, inner Asia,
and Southern North America show mostly no Mid-Holocene temperature maximum, but rather a Late
Holocene maximum. Records with an Early Holocene maximum dominate the north-central part of North
America and China, though these areas are characterized by high spatial variability. High ranges of
Holocene temperature variations (larger than 5°C) are found in mid-latitude Europe, Western Canada,
Southeastern US, and along the north Asian Pacific coast.
The averaged Northern Hemisphere north of 30°N time-series of all records that cover the entire
Holocene (Fig. 3) indicate that mean $T_{July}$ was lowest at the beginning of the Holocene (-0.7±0.2°C
compared to present), increased until 7 ka (+0.5±0.1°C compared to present), and slightly decreased
afterwards to reach modern temperatures. $T_{ann}$ was also lowest at the beginning of the Holocene (-
1.4±0.2°C compared to present) and reached its maximum of 0.3±0.2°C compared to present at 6.5 ka.
Finally, our revised global temperature curve includes all of our records and those of the Temp12k
dataset (Kaufman et al., 2020b) that cover the entire Holocene (in total, excluding duplicate pollen
records, 1098 records). It shows that mean $T_{ann}$ was lowest during the Early Holocene at 10.5 ka with a
-0.3±0.3°C anomaly relative to 1 ka and warmest at 6.6 ka with a warming of 0.3±0.3°C. For the Northern
Hemisphere extratropics (30-80°N), we find that mean $T_{ann}$ was lowest during the Early Holocene at
10.5 ka with a -0.3±0.1°C anomaly relative to 1 ka and warmest at 6.4 ka with a warming of 0.08±0.04°C.
The linear trends of all zonal means are significantly different (p < 0.01) for both $T_{July}$ (Table A2) and $T_{ann}$
(Table A3). While the uncertainty range is small in the mid-latitudes they are larger for the 30-40°N zonal
band ($T_{July}$) and especially for the polar region ($T_{July}$ and $T_{ann}$; Fig. A3). The linear trends for $T_{July}$ for all
continental means are significantly different, despite overlapping uncertainty ranges for several zonal
bands, e.g. 40-50°N and 50-60°N in Western North America (Fig. A4); 30-40°N and 50-60°N in Eastern
North America (Fig. A5), 30-40°N and 40-50°N, as well as 50-60°N and 60-70°N in Asia (Fig. A7). Large
uncertainty ranges can be found in the 30-40°N zonal band (Europe, Fig. A6) and the polar region
(Western North America, Fig. A4; Asia, Fig. A7). The linear trends for $T_{ann}$ reveal similarities between
the weighted means of Europe and Asia (Europe vs. Asia: p = 0.08; Asia vs. Europe: p = 0.9; Table A5).
For overlapping uncertainty ranges similar patterns compared to those of $T_{July}$ can be found, except for
Eastern North America, where the zonal means of 30-40°N and 50-60°N are very different to each other,
especially in the Early and Mid-Holocene (Fig. A5). Similar to $T_{July}$, the largest uncertainty ranges can
be found either in the 30-40°N or the 70-80°N zonal bands. For the weighted continental means the
uncertainty ranges of Western and Eastern North America show a strong overlap, i.e. the $T_{July}$ mean of
Eastern North America mirrors the weighted Northern Hemisphere $T_{July}$ mean. $T_{July}$ in Asia is lower
overall while in Europe it is higher overall than the Northern Hemispheric mean, but the uncertainty
range of both continental means are larger than those in North America (West and East) and the
Northern Hemisphere. For $T_{ann}$ the uncertainty ranges in all continents show a stronger overlap than for
$T_{July}$ with pronounced differences between the Western and the Eastern part of North America (Fig. A8).

**3.2 Spatio-temporal pattern of precipitation reconstructions**
Holocene mean $P_{ann}$ variations (as % of modern value) averaged across the Northern Hemisphere
extratropics have patterns that are mostly similar to $T_{ann}$ with $P_{ann}$ being lowest during the Early Holocene
(-11.6±2.8% at 11 ka compared to 1 ka) and increasing until 5.9 ka before becoming relatively stable
(Fig. 3).
In contrast to the averaged Northern Hemisphere pattern, the (sub-)continental precipitation patterns
differ from their respective temperature patterns. The mean precipitation time-series of Western North
America and Europe increases from the Early Holocene to the Late Holocene; averaged Eastern North
America precipitation increased until 6.5 ka and varies slightly around modern values from then; and
Asia shows a pronounced maximum between 7 and 5 ka.
Time-series maps of latitudinal means and differences (Fig. 4) reveal strong spatial patterns, particularly
for Asia. The latitudinal mean time-series in Asia show a strong increase toward the Mid-Holocene of
mostly >10%. After ca. 7 ka, certain differences exist: while the 70°N mean shows no clear further trend,
the other mean curves show a precipitation maximum which is at least 5% above the Late Holocene
minimum. Precipitation maxima (compared with the Late Holocene) are more pronounced and occur
later at lower latitudes. Furthermore, the 6-1 ka difference maps reveal that the Mid-Holocene moisture
maximum in subtropical Asia was most pronounced in East-central China with many records even
showing >=50% higher values at 6 ka compared to 1 ka (Fig. 4).
The Holocene precipitation increase in the other (sub-)continents is particularly strong in the 30-40°N
bands in subtropical Europe and mid-latitude North America with >13% and >20% precipitation increase,
respectively. In Europe and Western and Eastern North America the records from 70-80°N show an
Early Holocene precipitation maximum (particularly pronounced in Alaska), which is in contrast to the
trends in almost all other latitudinal bands.
Comparing the linear trends for all zonal means reveals significant differences in all zonal bands for
Europe and Eastern North America (p < 0.01). Similarities in the trends can be found in Western North
America (70-80°N vs. 30-40°N: p = 0.06) and especially in Asia, where several combinations of zonal
trends are not significantly different (i.e. 30-40°N vs. 40-50°N (p = 0.08) and 30-40°N vs. 70-80°N (p =
0.76)). For details, see Table A4. All trends in the continental precipitation means are found to be
different (p < 0.01; Table A5). The uncertainty ranges for all latitudinal means are small, except for the
70-80°N zonal band in the polar region (%$P_{ann}$; Fig. A3). In Western North America the zonal means of
50-60°N and 60-70°N show a strong overlap in their uncertainty ranges and the largest uncertainty range
can be found in the polar region (Fig. A4). In Europe and Asia, the mid-latitudes show the smallest
uncertainty ranges, while the southernmost and northernmost zonal bands have higher uncertainty
ranges (Fig. A6 and A7). Notable is the 40-50°N zonal band in Asia, which shows the highest uncertainty
range of all continental zonal bands, especially in the Mid-Holocene (Fig. A7). Compared to the Northern
Hemispheric mean, the continental %$P_{ann}$ mean of Eastern North America shows the smallest deviations,
although the continental mean only comprises the zonal bands between 30°N and 60°N. Precipitation
changes in Western North America are overall lower than the Northern Hemispheric mean, while the
precipitation changes in Asia are overall higher (Fig. A8).

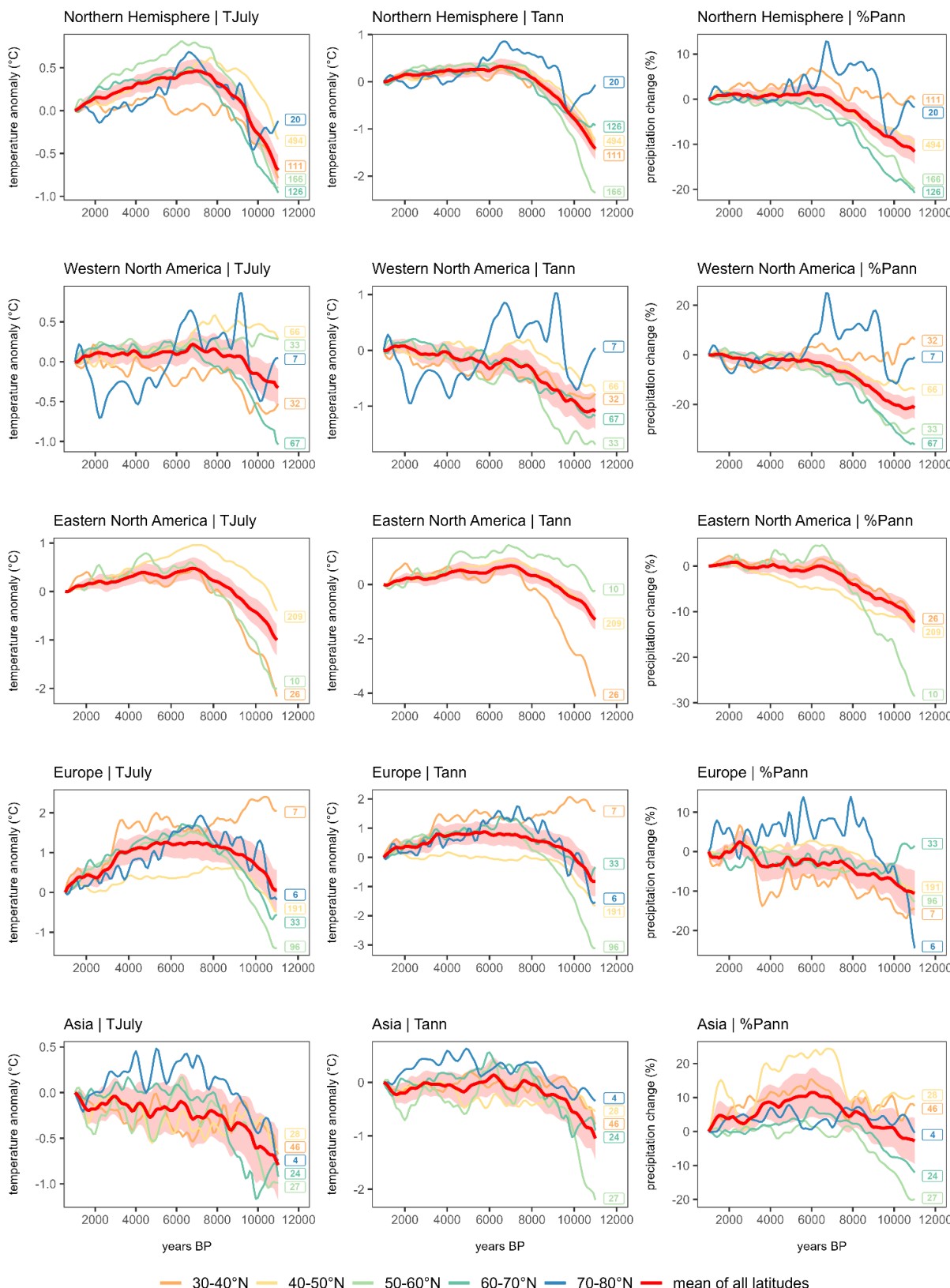


**Figure 3. Hemispheric, (sub-)continental, and zonal mean curves for $T_{July}$, $T_{ann}$, and $\%P_{ann}$ derived from pollen-based reconstruction with WA-PLS.** Curves from zonal bands that contain fewer than three grid cells were excluded. The shading corresponds to the latitude-weighted standard error of the latitude-weighted mean. Labels in corresponding colors indicate the number of grid boxes that contributed to each latitudinal curve.

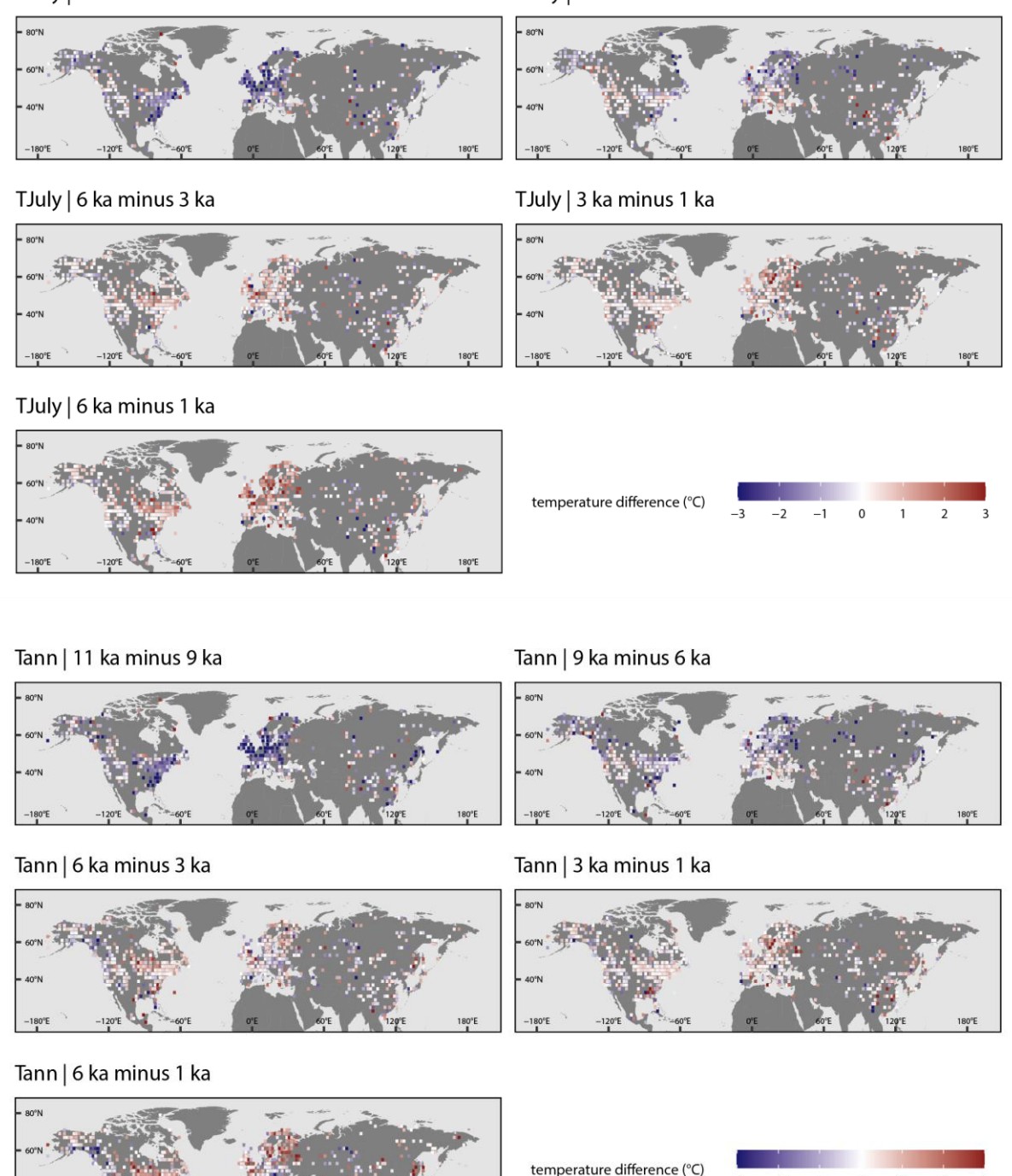

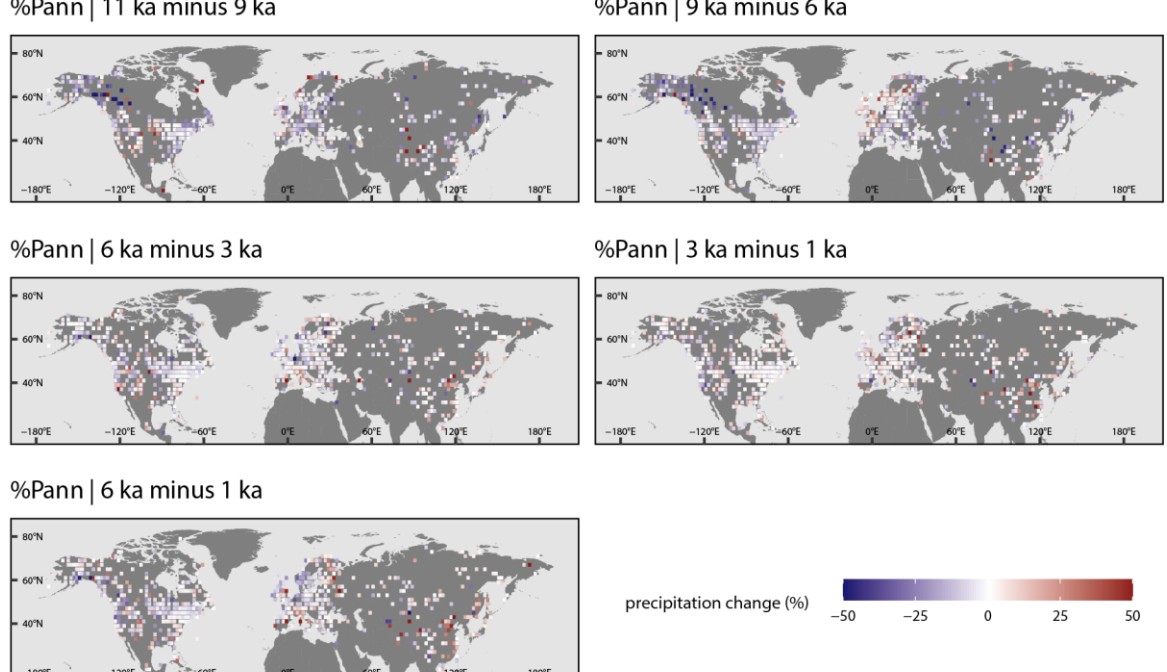

336

**Figure 4. Difference maps of $T_{July}$, $T_{ann}$ (°C), and $P_{ann}$ (as % of the value of the younger time-slice)**
**between selected time-slices.** Color code for values outside the range were restricted to range maxima.
A list with the entire value range and the proportions of values that fall within the restricted range are
presented in Table A1. Maps are gridded values averaging the values of records from within the 2°x2°
grid cell.

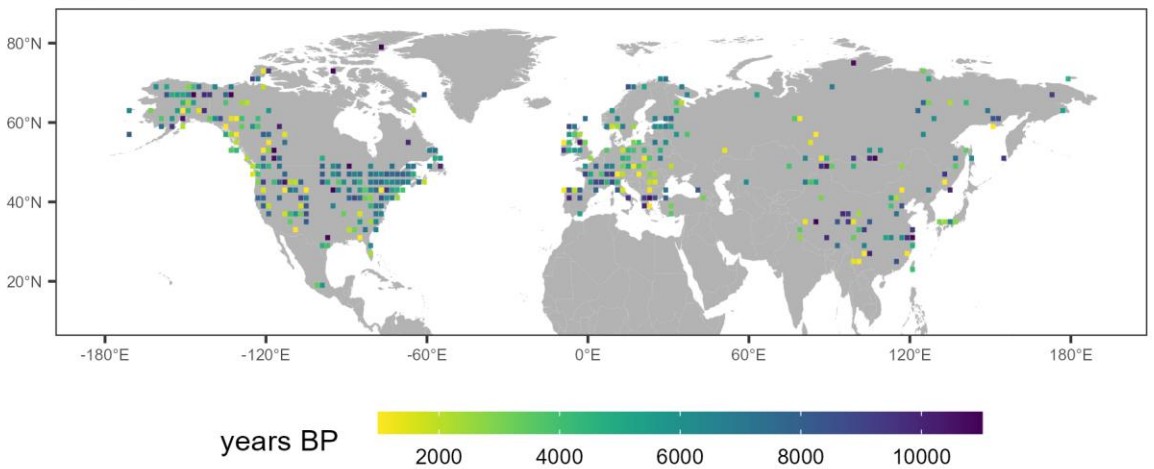

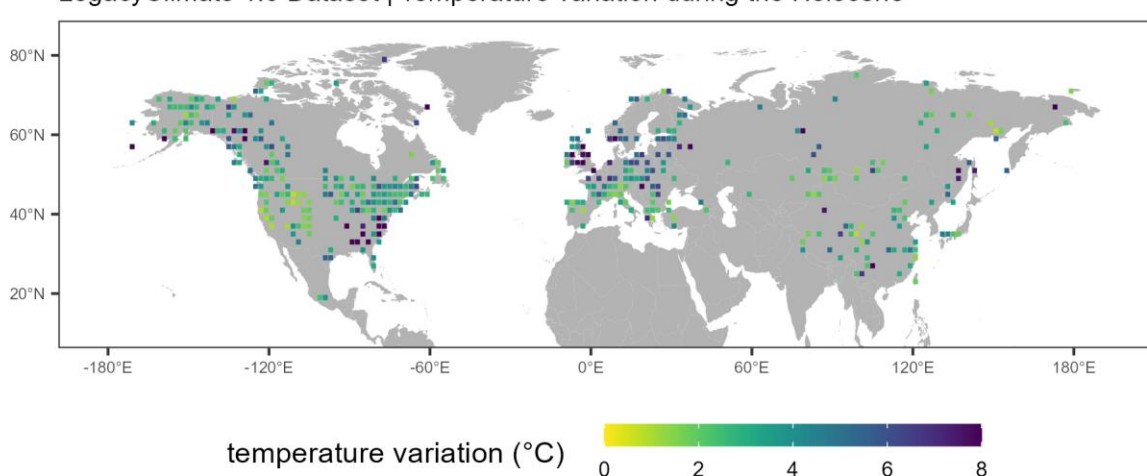

342

**Figure 5. Maps indicating the timing of the $T_{ann}$ maximum (top) and the range of $T_{ann}$ variation during the Holocene (11-1 ka, bottom).** Each 2°x2° grid cell contains the averaged values of all records located within one grid cell. For each grid cell, the $T_{ann}$ variation was determined as the range between minimum and maximum $T_{ann}$ anomalies. The $T_{ann}$ Holocene temperature maximum is the timing of the anomaly maximum. Color code for values outside the range were restricted to range maxima.

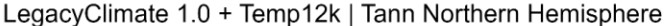

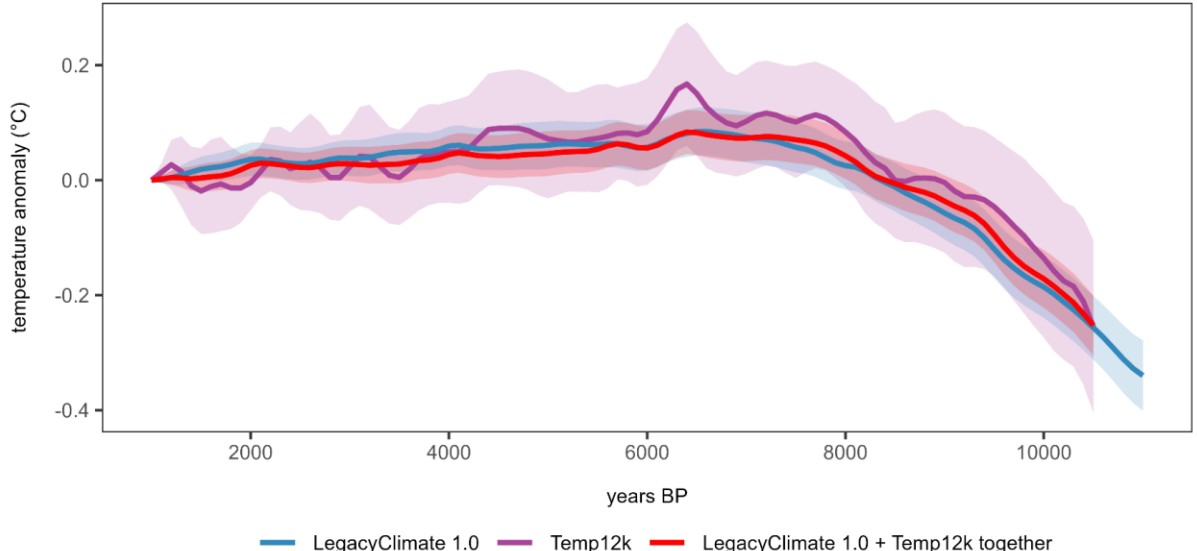

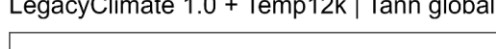

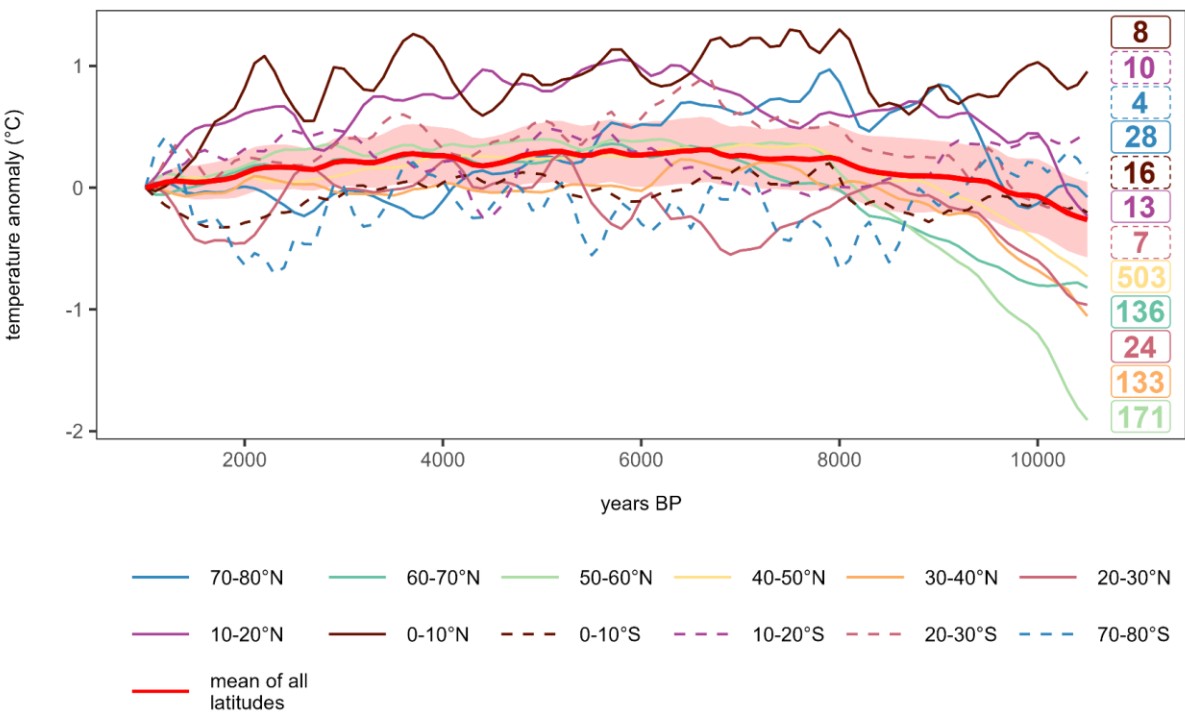

348

**Figure 6. Mean curves for temperature. (**top) Northern Hemisphere weighted means with shaded
weighted standard error (no curves for latitudes): LegacyClimate 1.0 (n=957; blue), Temp12k dataset
(n=272, see methods for record filter; purple,), LegacyClimate 1.0 + Temp12k mean (n=1098; red);
(bottom) LegacyClimate 1.0 + Temp12k global mean with latitudinal means. Labels in corresponding
colors indicate the number of grid boxes that contributed to each latitudinal curve.

354

355

356

**4 Discussion**

**4.1 Spatial temperature pattern (in light of the global Holocene temperature curve)**

The general pattern of the LegacyClimate 1.0 mean annual temperature curve of the Northern Hemisphere extratropics agrees with those of previous investigations (Marcott et al., 2013; Kaufman et al., 2020b; Kaufman and Broadman, 2023) including a cold Early Holocene, a temperature maximum during the Early to Mid-Holocene, and a slight cooling towards the present-day (Fig. 2; Fig. A8). Orbital forcings are assumed to have an important influence on the trends in the global mean temperatures, which led to feedback mechanisms like decreased polar sea ice or shifted vegetation ranges and thus to increased temperatures during the Mid-Holocene (Kaufman and Broadman, 2023). Subsequently, changes in solar irradiance, an increasing albedo due to land-cover changes and increasing volcanic activity probably contributed to a global cooling during the Late Holocene (Kaufman and Broadman, 2023). Both our LegacyClimate 1.0 and the Temp12k mean temperature curves increase from the Early Holocene to the Mid-Holocene by about 0.4°C when the same stacking approach is applied. However, the LegacyClimate 1.0 stack shows only a minimal temperature decline between the early Mid-Holocene maximum and the Late Holocene minimum of ~0.08°C compared to ~0.17°C in the Temp12k stack. We suggest two probable reasons for this finding: 1) a more complete spatial and temporal representativeness of the dataset, and 2) a unique methodology to reconstruct a small set of climate variables from pollen data.

First, our mean annual temperature curve includes about four times as many records as the Temp12k dataset (957 records in the LegacyClimate 1.0 dataset vs. 272 records in the Temp12k dataset, Kaufman et al. 2020b; Fig. 1). In particular, Asia is represented by substantially more records in the combined dataset. Our temperature reconstruction from Asia shows an average trend that differs from the overall Northern Hemisphere trend as it has no pronounced Holocene temperature maximum (Fig. A8; Table A6). This is particularly true for Asian $T_{ann}$ records south of 50°N and $T_{July}$ records south of 60°N. This feature has not been recognized so far, likely because Asian temperature reconstructions are mostly lacking in previous compilations (e.g., Marcott et al., 2013; Marsicek et al., 2018; Routson et al., 2019; Kaufman et al., 2020b). Even if the Mid- to Late Holocene cooling trend observed in Asia north of 60°N (Fig. 2) agrees with the proposed Neoglacial (sub-)arctic-wide Holocene cooling, the amount of cooling of <0.5°C is low compared to the cooling observed in other regions (e.g., in Europe where an average cooling of ~1.5°C has been reconstructed; McKay et al., 2018; Fig. 2). As with the differences between Eastern and Western Eurasia, we find a difference between Eastern and Western North America. In particular, we can identify a circum-North Atlantic pattern with a strong Early Holocene increase, a pronounced Mid-Holocene maximum and strong temperature range, and a circum-North Pacific pattern with an overall weak change. This is likely related to the impact of the decaying Laurentide ice-sheet on the North Atlantic which was probably a stronger driver of Early to id-Holocene temperature change than insolation (Renssen et al., 2009; Renssen et al., 2012; Zhang et al., 2016).

Even if this study shows a less pronounced Holocene temperature maximum, the problem remains that this does not align with the overall Holocene increase in the mean global (and Northern Hemisphere) temperature revealed by Earth System Models. Our study points to a strong regionalization of Holocene

temperature trends and range of variation in the Northern Hemisphere extratropics, which was also reported in recent studies (e.g. Kaufman et al., 2020b; Osman et al., 2021; Cartapanis et al., 2022). This somehow contradicts the ´Holocene conundrum´ concept which tackled Holocene temperature change mainly by analyzing the global mean and understanding the differences between proxy-based and simulated reconstructions. However, the conundrum debate has since progressed and recent studies hint at discrepancies in data-model comparisons due to spatiotemporal dynamics related to heterogeneous responses to climate forcing and feedbacks (e.g., the timing of a Holocene thermal maximum between reconstructions from continental and from marine proxy records; Cartapanis et al., 2022). Our finding is in line with recent modeling approaches, which also yield strong regional differences in temperature developments (Bader et al., 2020) allowing for a regional comparison. Recent paleo-data assimilation approaches based on marine temperature reconstructions reveal peculiarities of spatial averaging as one reason for the model-data mismatch (Osman et al., 2021). The error is most pronounced where the number of included records is small. This stresses the importance of good spatial coverage of the records used for the assessment of the mean temperature trend. Including terrestrial reconstructions is crucial. Compared with previous syntheses of terrestrial records, our compilation is notable for its higher record density in Asia, a region for which Earth System Models show diverging past climate changes, highly sensitive to boundary conditions and forcing (Bakker et al. 2020; Brierley et al., 2020; Lohmann et al. 2021). Therefore, our reconstruction makes a decisive contribution to locating and clarifying the model-data mismatch in the Northern Hemisphere extratropics. From a proxy perspective, future targets of synthesis studies should focus on the Southern Hemisphere and poorly covered areas in Central Asia and Siberia.

Second, standardized methodologies may have contributed to the observed differences between the LegacyClimate 1.0 mean $T_{ann}$ curve and the Temp12k curve. Our $T_{ann}$ reconstruction only includes records of mean annual temperature while the Temp12k product mixes reconstructions of seasonal temperature (mostly $T_{July}$) if $T_{ann}$ is not available from a site. This assumption of equivalence between annual and summer temperature at any given site can impact the trend and amplitude of the stacks. A seasonal bias in the reconstructions may originate from a real, larger Holocene range of summer temperature variations (Bova et al., 2021) or is an artefact introduced by having a larger $T_{July}$ range covered by the calibration datasets compared with $T_{ann}$ which is, however, not the case in our calibration sets.

Our pollen-based reconstructions are all performed with WA-PLS, which is known to produce smaller climate amplitudes than MAT (a likewise commonly used method) because it is less sensitive to extreme climate values in the modern pollen dataset (Birks and Simpson 2013; Cao et al., 2017; Nolan et al. 2019). Furthermore, by using a standard area size for our modern pollen datasets, we may have stabilized the regional reconstructions, that is, equalized the amplitude as the source areas represent rather similar biogeographical and climate ranges. Finally, our reconstructions include only records that cover the entire Holocene period (11-1 ka) and not just parts of it. Hence, all time-slices have a similar spatial coverage and the temporal pattern is not biased by regions where archives are only available in certain periods (e.g., the Late Holocene peatland establishment).

As with all applications of taxa-based transfer functions to fossil records, we assume that both modern
and past taxa assemblages (in our case, vegetation) are in equilibrium with climate, and that the
relationships inferred from modern data do not change throughout the Holocene (Birks et al., 2010;
Chevalier et al., 2020) and that the modern pollen assemblages are not heavily biased by human impact.
Differences in global boundary conditions during the Early to id-Holocene (e.g., lower atmospheric $CO_2$
concentration, different seasonal insolation) however, may have modified these relationships, which
could have also dampened the reconstructed amplitudes. Also, vegetation response to climate change
may be involve lags (see the ongoing discussion about the so-called ´forest conundrum´, i.e., the
observation that observed forest maximum lags the simulated temperature maximum; Dallmeyer et al.,
2022) and depends on the initial conditions such as the distribution of refugia during the Last Glacial
(Herzschuh et al., 2016; 2020). Furthermore, there are areas, especially the densely settled regions in
Europe and Southeastern Asia, that are affected by human activities throughout the Holocene due to
intense animal husbandry, as inferred from the abundance of Plantaginaceae and Rumex as indicators
of grazing (Herzschuh et al., 2022a), or due to industrialization since the second half of the 19th century.
This probably led to extinction events, especially for disturbance-dependent taxa and contributed to
gaps within the potential bioclimatic space of taxa that form natural communities (Zanon et al., 2018).
The absolute effect of these biases is hard to quantify (but see Cleator et al., 2020), and many
comparative, multi-proxy Holocene studies have shown that pollen-based reconstructions are as reliable
as any other proxy (Kaufmann et al., 2020a; Dugerdil et al., 2021). In contrast, one advantage of single
proxy studies is that any biases will affect all the records similarly. As such, even if the actual amplitude
of our regional and global stacks might be dampened, the trends and spatial patterns shared by the data
are likely to remain correct.

## 4.2 Spatio-temporal precipitation pattern

Our analyses of the Holocene spatio-temporal precipitation pattern fill a research gap, as syntheses of
proxy-based precipitation change on a hemispheric scale during the Holocene are still lacking. Regional
syntheses are available for Europe (Mauri et al., 2014 and 2015), North America (Ladd et al., 2015;
Routson et al., 2021), and Eastern Asia (Herzschuh et al., 2019). Interestingly, we observed a similar
pattern for Northern Hemisphere-wide averaged Holocene trends of $P_{ann}$ and $T_{ann}$, but differences
among corresponding $P_{ann}$ and $T_{ann}$ curves at (sub-)continental and latitudinal scales, e.g., in Asia, where
the $P_{ann}$ means are overall higher than the Northern Hemispheric means while the $T_{ann}$ means are overall
lower since ~ 9 ka (Fig. A8), or for the 30-40°N zonal band, where $T_{ann}$ shows an Early to Mid-Holocene
warming while no trend in the $P_{ann}$ means could be found for this time period (Fig. A3).
This regional heterogeneity with respect to the precipitation trend (i.e., significantly different trends for
the Northern Hemisphere except for some regions in Asia, Table A4, Fig. A8) is also seen in recent
Earth System Model simulations for the last 8000 years (Mauri et al., 2014; Dallmeyer et al., 2021).
Although the simulated pattern does not exactly match our reconstructions, they share many similar
structures such as high precipitation in the Early and Mid-Holocene in East Asia (Fig. 4). For this region,
our reconstruction shows the strongest Mid- to Late Holocene precipitation decline worldwide, reflecting
the weakening of the East Asian Summer Monsoon (EASM) in response to the decrease in summer
insolation. This trend in moisture has been confirmed by earlier qualitative and quantitative proxy
syntheses and modeling studies (Wang et al., 2010; Zheng et al., 2013; Liu et al., 2014a; Herzschuh et
al., 2019).
In contrast, many Central Asian sites show low Early-Holocene precipitation levels (Fig. 4). This anti-
phase relationship in EASM to Central Asian moisture change is in line with earlier studies (Jin et al.,
2012; Chen et al., 2019; Herzschuh et al., 2019; Zhang et al., 2021). The causal mechanisms are still
debated. Among other reasons, precipitation-evaporation effects (Herzschuh et al., 2004; Zhang et al.,
2011; Kubota et al., 2015), transcending air mass related to the Rodwell-Hoskins response to
monsoonal heating (Herzschuh et al., 2004; Wang et al., 2017), effects from winter precipitation (Li et
al., 2020), and translocation of the westerly jetstream (Herzschuh et al., 2019) may contribute to the
anti-phased precipitation change.
Arctic warming mechanistically should be linked with wetting in the Arctic due to high hydrological
sensitivities (Trenberth, 2011). Such a pattern is, for example, obvious for Early to id-Holocene climate
change in most records from Alaska. Interestingly, several records from the northern Arctic coastal
region in Russia, northern Norway and Canada show a wet Early Holocene, which is also observed in
simulations (Dallmeyer et al., 2021).
Contrasting the trend in the East Asian monsoon region (Fig. 2; Fig. A7), annual precipitation increases
in mid-latitude Europe during the Holocene according to our reconstructions (Fig. 2; Fig. A6). Routson
et al. (2019) propose a circum-hemispheric mid-latitudinal rise of moisture levels over the Holocene
based on a semi-quantitative dataset that is strongly concentrated around the circum-Atlantic region.
They relate the decreased net precipitation to the weakened Early Holocene latitudinal temperature
gradient. Due to polar amplification, the arctic regions experienced a stronger warming in the climate
compared to the equatorial region, which is also supported by our dataset. However, we also see in our
reconstructions that this view is too general, but it may explain the precipitation response in Europe as
the weakening of the latitudinal temperature gradient is particularly pronounced in Europe in our
reconstructions. This change in temperature pattern is probably a result of a dampening in the cyclonic
activity along the weaker westerly jet (Chang et al., 2002; Routson et al., 2019; Xu et al., 2020), bearing
less precipitation during the Early Holocene compared to modern conditions. With the strengthening of
the latitudinal temperature gradient towards the Late Holocene, cyclonic activity enhances, leading to
an increase of precipitation over the Holocene.
According to our reconstructions, the precipitation trend in Eastern and Western North America strongly
differs (p < 0.01; Table A5; Fig. A3). While in the Eastern part the mean precipitation level is relatively
stable in all latitudinal bands, except the 50-60°N zonal band, over the Holocene (Fig. A5), precipitation
strongly increases on average in the Western part (Fig. A4), driven by a precipitation rise in the mid-
latitudes (40-70°N). In the polar regions and south of 40°N, precipitation declines from the Mid-Holocene
(Fig. 4; Fig. A4). The latter may be related to a decrease in the North American monsoon intensity, in
line with the orbital monsoon hypothesis (Kutzbach, 1981; Harrison et al., 2003). In the polar region,
modeling studies report northward shifted storm tracks coinciding with a northward replaced upper

tropospheric jetstream in the Mid-Holocene compared to the Late Holocene, promoting precipitation in the arctic region and decreasing precipitation at mid-latitudes (Zhou et al., 2020; Dallmeyer et al., 2021). With the southward shift of the polar jet during the Holocene, precipitation decreased in the high northern latitudes in North America and increased further south (Liu et al., 2014b).

The rise in moisture levels across the North American continental interior over the course of the Holocene has been proposed before (Grimm et al., 2001; Zhou et al., 2020; Dallmeyer et al., 2021) but has not yet been quantified with continental-wide proxy-data. The main drivers of this trend are still being debated: besides shifts in the westerly wind circulation (Seager et al., 2014), weakening subsidence caused by teleconnection with the weakening Northern Hemispheric monsoon systems (Harrison et al., 2003; Dallmeyer et al., 2021), reorganization of the atmospheric circulation around the Bermuda high (Grimm et al., 2001), and changes in the sea-surface temperature pattern (Shin et al., 2006) may contribute to an increase in precipitation over the Holocene.

Reconstructing temperature and precipitation from a single dataset implies that they are both important in defining the presence and/or abundance of specific pollen taxa (Salonen et al., 2019). This hypothesis cannot be tested but to some extent has been assessed by several analyses (Juggins, 2013). The WA-PLS reconstruction was also applied with tailored modern calibration sets (i.e., selecting samples so that the correlation between temperature and precipitation in the calibration dataset is reduced). The finding that the reconstructions were generally very similar between those using the full and those using the tailored modern datasets can be taken as an indication that co-variation is not a major issue in these reconstructions (Herzschuh et al., 2022a). This conclusion is also supported by the fact that $T_{ann}$ and $P_{ann}$ records that pass the reconstruction significance test when the impact of the other variable is partialled out (Telford and Birks, 2011), are almost evenly distributed over the Northern Hemisphere records (Herzschuh et al., 2022a). This is also confirmed by the visual inspection of the regional reconstructions in Fig. 3, where we cannot detect correlations between variables within latitudinal zones, as would be expected from dependent reconstructions. This suggests that our reconstructions do reflect distinctive trends from the pollen data.

## 5 Conclusions

We investigated Holocene time-series of $T_{July}$, $T_{ann}$, and $P_{ann}$ for the Northern Hemisphere extratropics making use of 2593 pollen-based reconstructions (LegacyClimate 1.0). Compared with previous datasets, we include many more records, particularly from Asia. We present mean curves obtained with the same method for the Northern Hemisphere, the (sub-)continents (Asia, Europe, Eastern North America, Western North America), and regional zones (i.e., 10° latitudinal bands for (sub-)continents) as well as Northern Hemisphere gridded data for selected time-slices.

Our results indicate that Holocene climate change shows unique regional patterns. The concept of a Mid-Holocene temperature maximum only applies mainly to the mid and high northern latitudes in the circum-North Atlantic region while records from mid-latitude Asia, Western North America, and all subtropical areas do not fit into this concept but mostly show an overall Holocene increase or other

patterns. As such, the ´Holocene conundrum´, originally proposed as a global feature, may instead apply
to a restricted region.
The precipitation trend is roughly similar to the temperature trend at the hemispheric scale, in particular
with respect to the strong increase from the Early to Mid-Holocene. At the regional scale, the
precipitation trends differ from each other and also from the regional temperature trends. The 40-50°
latitudinal band in Asia shows the most pronounced Mid-Holocene precipitation maxima while many
regions show increasing Holocene trends including most of Europe and Western North America. We
relate these differences to regionally specific circulation mechanisms and their specific relationships
with temperature changes.
Given a background of strong regional heterogeneity, the calculation of global or hemispheric means
might generally lead to misleading concepts but the focus should be on understanding the spatio-
temporal patterns requiring spatially dense proxy-datasets for comparison with simulations.

**6 Data Availability**
The compilation of reconstructed $T_{July}$, $T_{ann}$, and $P_{ann}$, is open access and available at PANGAEA
(https://doi.org/10.1594/PANGAEA.930512; in the "Other version" section). The dataset files are stored
in machine-readable data format (.CSV), which are already separated into Western North America,
Eastern North America, Europe, and Asia for easy access and use.

**Author contributions.** UH designed the study. The analyses were led by UH and implemented by TB.
UH guided the interpretation of the results and collected detailed comments from AD, MC, OP, CL, and
RH. All co-authors commented on the initial version of the manuscript.

**Competing interests.** The authors declare that they have no conflict of interest.

**Acknowledgements.** We would like to express our gratitude to all the palynologists and geologists who,
either directly or indirectly by providing their work to the Neotoma Paleoecology Database, contributed
pollen data and chronologies to the dataset. The work of data contributors, data stewards, and the
Neotoma community is gratefully acknowledged. We also thank Cathy Jenks for language editing.

**Financial support.** This research has been supported by the European Research Council (ERC Glacial
Legacy 772852 to UH) and the PalMod Initiative (01LP1510C to UH). TB, MC, and AD are supported
by the German Federal Ministry of Education and Research (BMBF) as a Research for Sustainability
initiative (FONA; https://www.fona.de/en) through the PalMod Phase II project (grant no. FKZ:
01LP1926D and 01LP1920A). CL holds a scholarship from the Chinese Scholarship Council (grant no.
201908130165). NR work was supported by the Russian Science Foundation (Grant No. 20-17-00110).

**Appendix**


**Table A1.** Range of values in the difference maps (Fig. 4) and proportion of values that fall within a
restricted range of -3 to +3 °C for temperature and -50% to 50% for precipitation change.

| | $T_{July}$ | | $T_{ann}$ | | $P_{ann}$ | |
|---|---|---|---|---|---|---|
| | Value range | % within restricted range | Value range | % within restricted range | Value range | % within restricted range |
| **11-9 ka** | -12.3°C to +8.2°C | 87.8 % | -20.0°C to +6.0°C | 79.7 % | -131.7% to +151.3% | 96.9 % |
| **9-6 ka** | -6.1°C to +16.4°C | 95.8 % | -8.9°C to +12.0°C | 92.9 % | -81.4% to +103.9% | 98.4 % |
| **6-3 ka** | -8.2°C to +6.4°C | 98.1 % | -8.0°C to +7.9°C | 96.5 % | -175.1% to +423.6% | 98.8 % |
| **3-1 ka** | -10.1°C to +4.6°C | 98.2 % | -11.0°C to +10.1°C | 97.2 % | -1157.4% to +90.7% | 99.0 % |
| **6-1 ka** | -9.6°C to +6.5°C | 94.9 % | -8.9°C to +9.0°C | 93.6 % | -67.6% to +694.3% | 98.2 % |

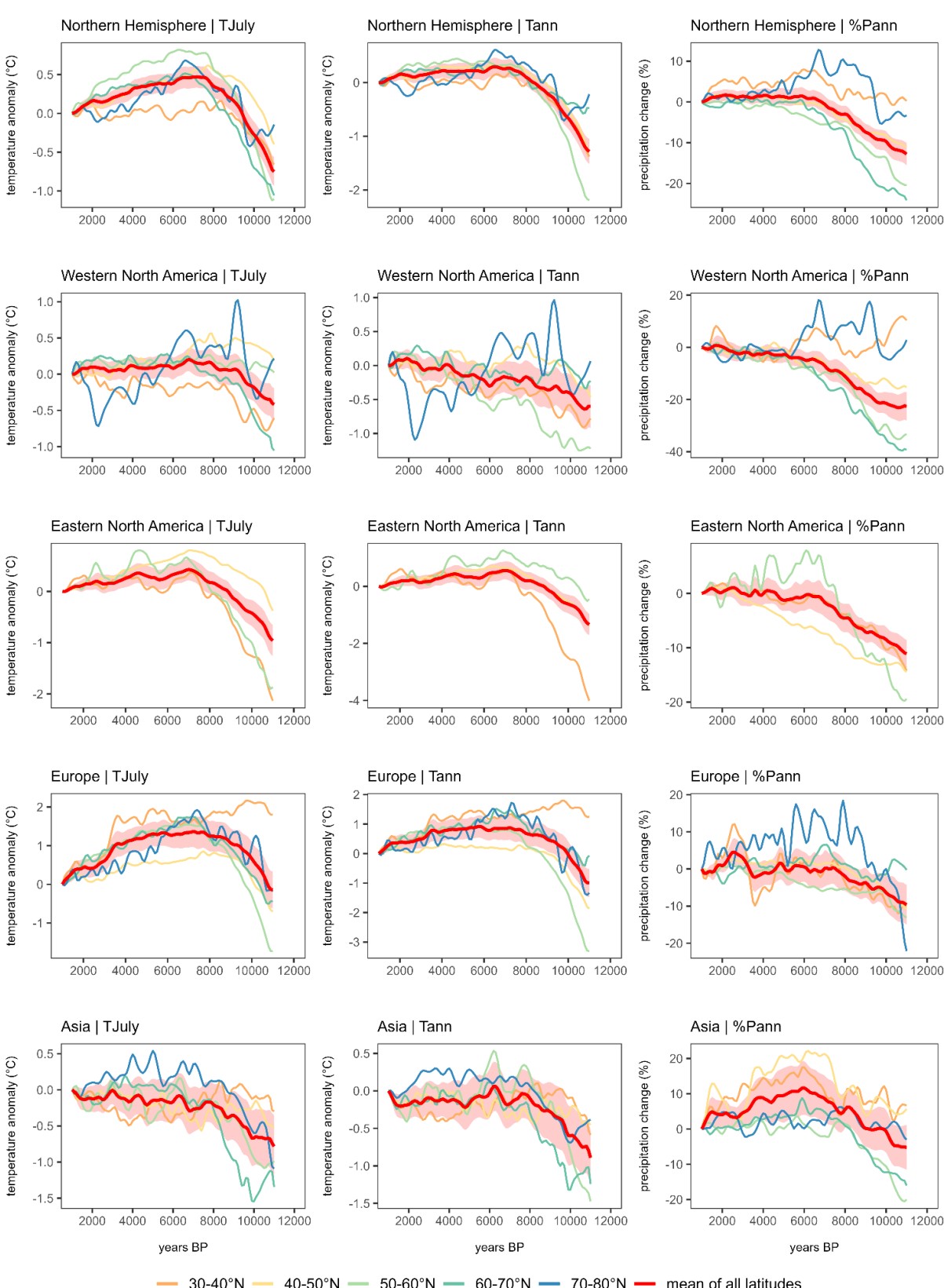


**Figure A1: Hemispheric, continental, and latitudinal mean curves for T$_{July}$, T$_{ann}$, and P$_{ann}$ derived from pollen-based reconstruction with WA-PLS_tailored.** Latitudinal bands that contain fewer than three grid cells are not shown. The shading corresponds to the latitude-weighted standard error of the latitude-weighted mean.

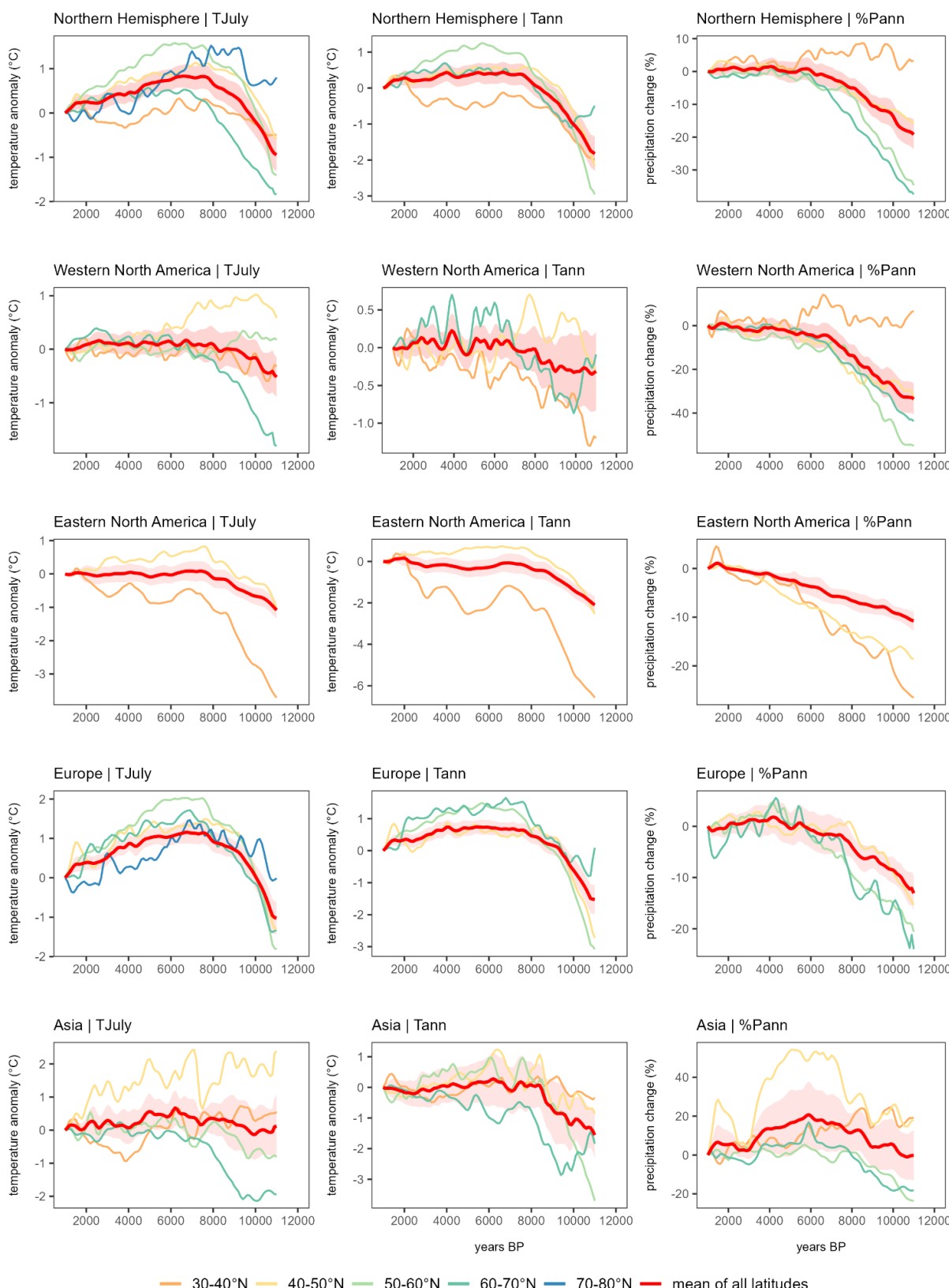

597

**Figure A2: Hemispheric, continental, and latitudinal mean curves for $T_{July}$, $T_{ann}$, and $P_{ann}$ derived from pollen-based reconstruction with WA-PLS_tailored with significant records (p < 0.2).** Latitudinal bands that contain fewer than three grid cells are not shown. The shading corresponds to the latitude-weighted standard error of the latitude-weighted mean.

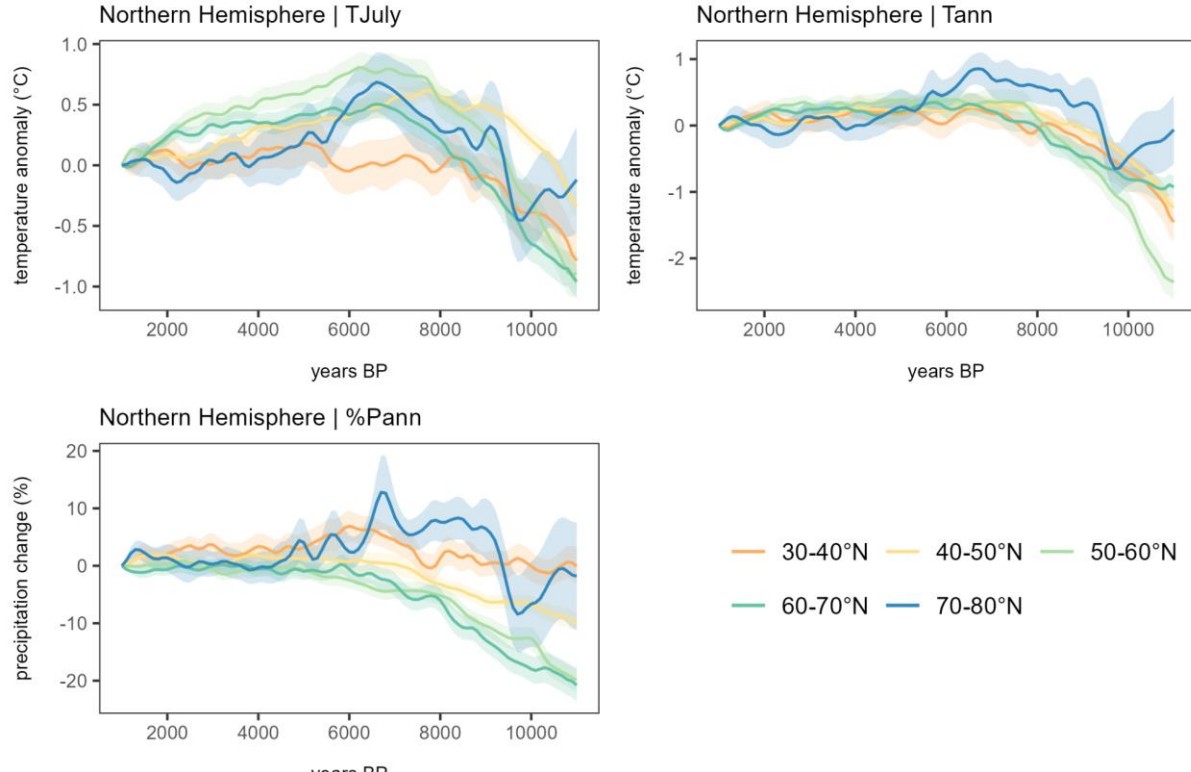


**Figure A3: Northern Hemispheric latitudinal mean curves with shaded standard errors for $T_{July}$, $T_{ann}$, and %$P_{ann}$ derived from pollen-based reconstruction with WA-PLS (latitudinal bands that contain fewer than three grid cells are not shown).**


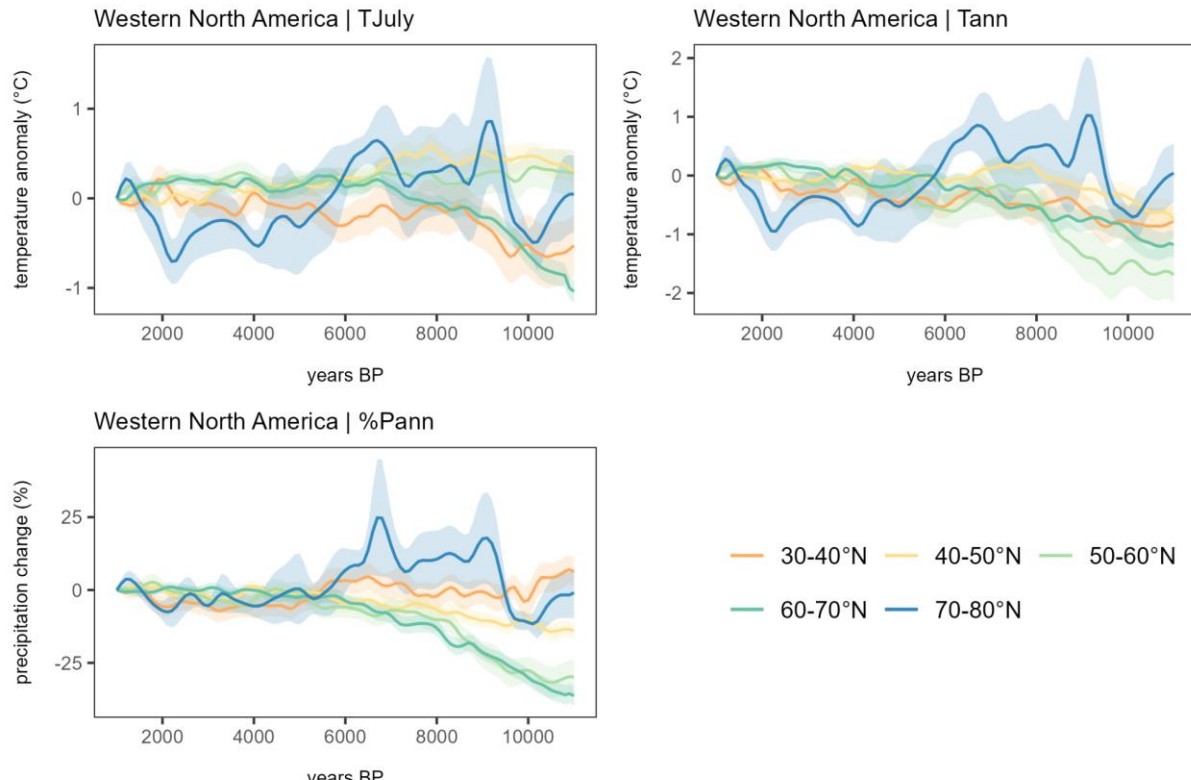


**Figure A4: Western North American latitudinal mean curves with shaded standard errors for T$_{July}$,**
**T$_{ann}$, and %P$_{ann}$ derived from pollen-based reconstruction with WA-PLS (latitudinal bands that**
**contain fewer than three grid cells are not shown).**

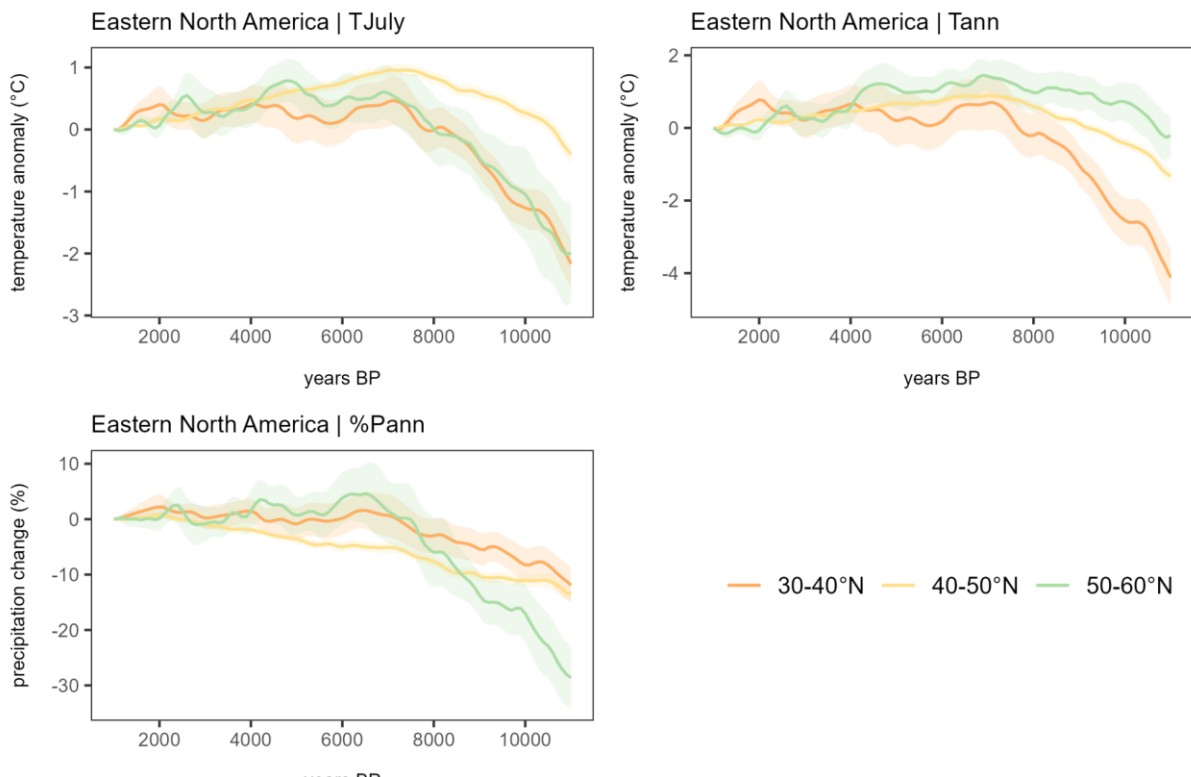


**Figure A5: Eastern North American latitudinal mean curves with shaded standard errors for T$_{July}$,**
**T$_{ann}$, and %P$_{ann}$ derived from pollen-based reconstruction with WA-PLS (latitudinal bands that**
**contain fewer than three grid cells are not shown).**

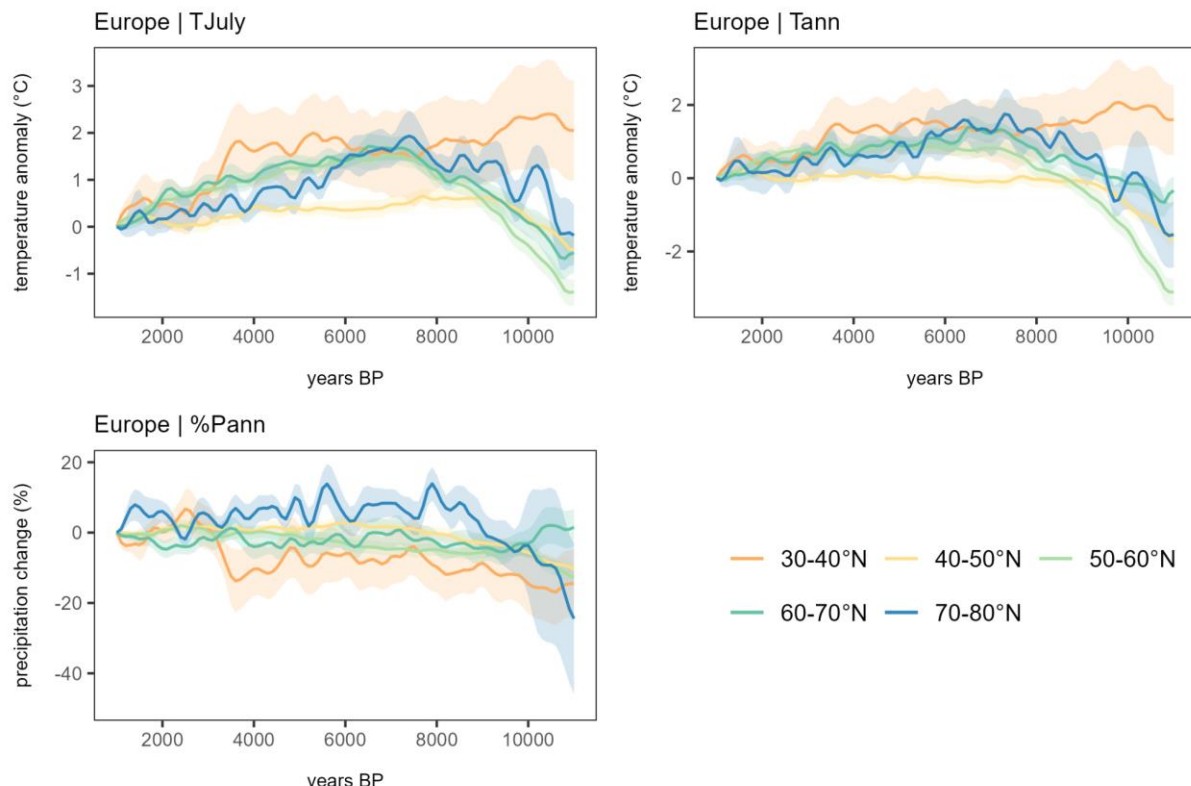


**Figure A6: European latitudinal mean curves with shaded standard errors for $T_{July}$, $T_{ann}$,**

**and %$P_{ann}$ derived from pollen-based reconstruction with WA-PLS (latitudinal bands that contain**

**fewer than three grid cells are not shown).**


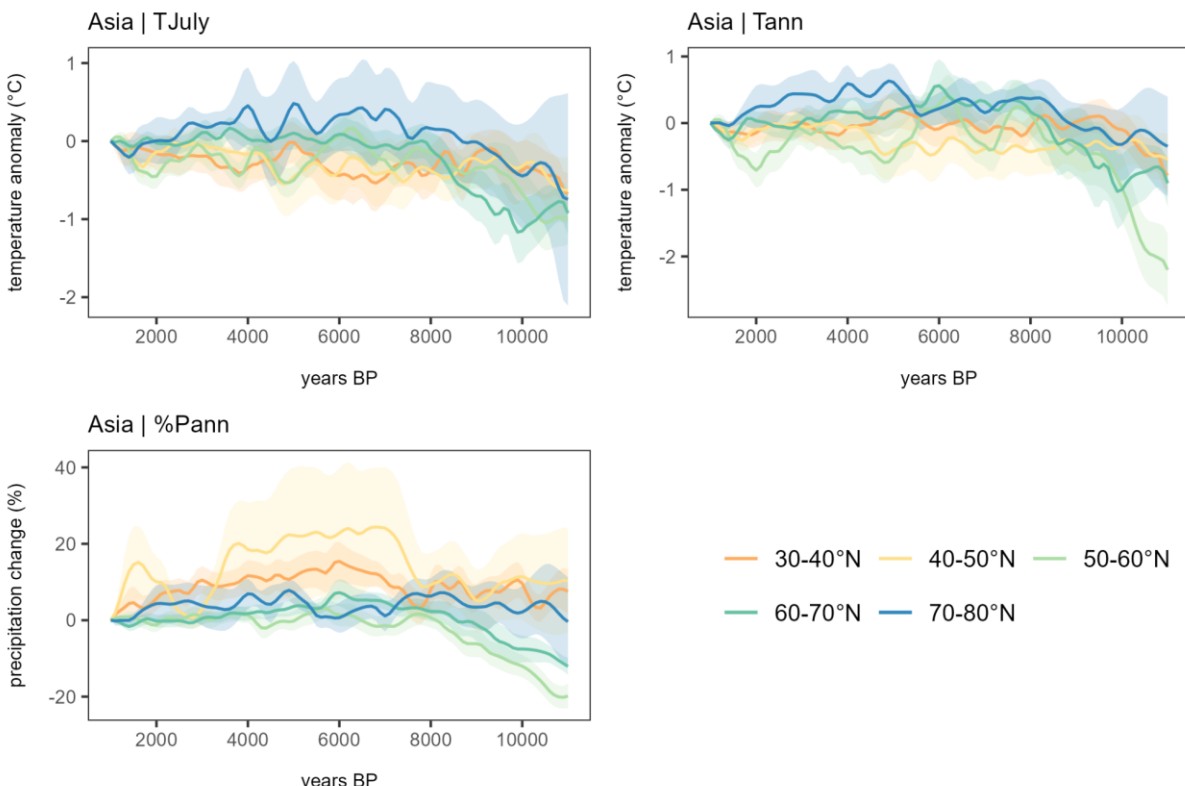


**Figure A7: Asian latitudinal mean curves with shaded standard errors for $T_{July}$, $T_{ann}$, and $\%P_{ann}$**
**derived from pollen-based reconstruction with WA-PLS (latitudinal bands that contain fewer than**
**three grid cells are not shown).**


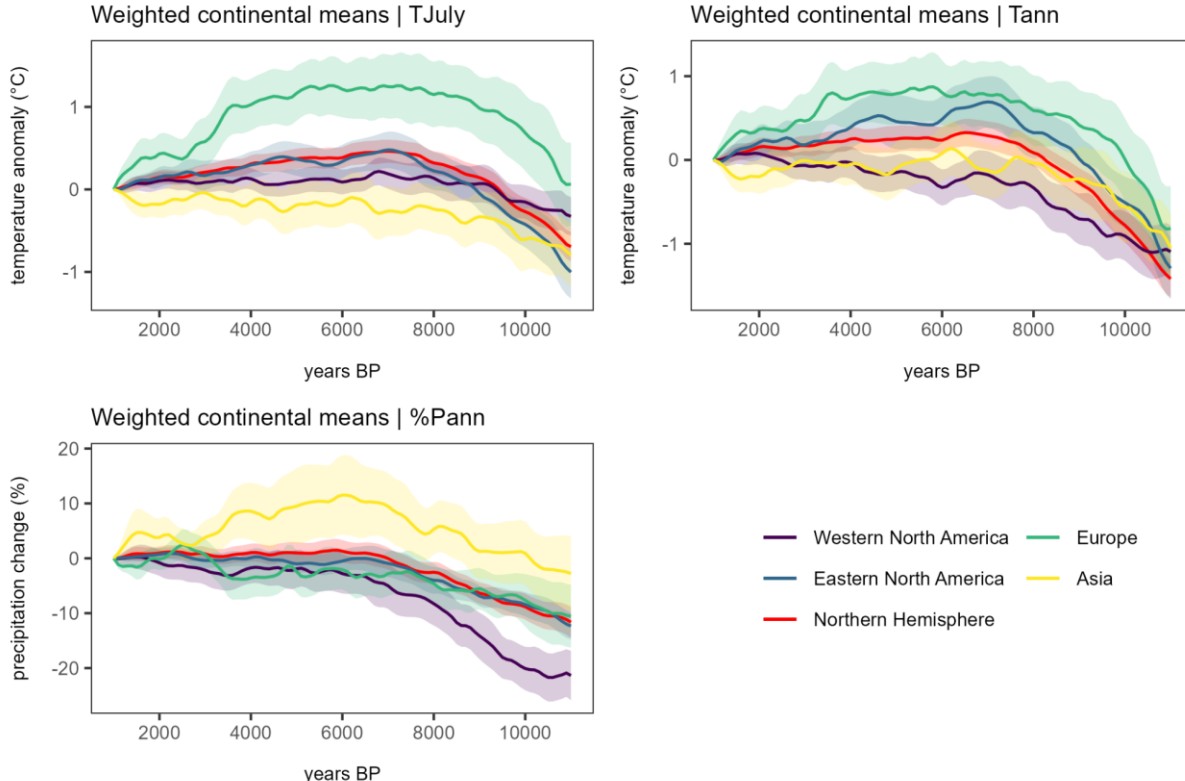


**Figure A8: Weighted continental means with shaded standard errors for $T_{July}$, $T_{ann}$, and $\%P_{ann}$**
**derived from pollen-based reconstruction with WA-PLS.**

















**Table A2.** Significance values for zonal linear trends derived from a Monte-Carlo test comparison for
mean July temperatures ($T_{July}$).

| | | 30-40°N | 40-50°N | 50-60°N | 60-70°N | 70-80°N |
|---|---|---|---|---|---|---|
| **Western North America** | 30-40°N | | p < 0.01 | p < 0.01 | p < 0.01 | p < 0.01 |
| | 40-50°N | p < 0.01 | | p < 0.01 | p < 0.01 | p < 0.01 |
| | 50-60°N | p < 0.01 | p < 0.01 | | p < 0.01 | p < 0.01 |
| | 60-70°N | p < 0.01 | p < 0.01 | p < 0.01 | | p < 0.01 |
| | 70-80°N | p < 0.01 | p < 0.01 | p < 0.01 | p < 0.01 | |
| **Eastern North America** | 30-40°N | | p < 0.01 | p < 0.01 | p < 0.01 | p < 0.01 |
| | 40-50°N | p < 0.01 | | p < 0.01 | p < 0.01 | p < 0.01 |
| | 50-60°N | p < 0.01 | p < 0.01 | | p < 0.01 | p < 0.01 |
| | 60-70°N | p < 0.01 | p < 0.01 | p < 0.01 | | p < 0.01 |
| | 70-80°N | p < 0.01 | p < 0.01 | p < 0.01 | p < 0.01 | |
| **Europe** | 30-40°N | | p < 0.01 | p < 0.01 | p < 0.01 | p < 0.01 |
| | 40-50°N | p < 0.01 | | p < 0.01 | p < 0.01 | p < 0.01 |
| | 50-60°N | p < 0.01 | p < 0.01 | | p < 0.01 | p < 0.01 |
| | 60-70°N | p < 0.01 | p < 0.01 | p < 0.01 | | p < 0.01 |
| | 70-80°N | p < 0.01 | p < 0.01 | p < 0.01 | p < 0.01 | |
| **Asia** | 30-40°N | | p < 0.01 | p < 0.01 | p < 0.01 | p < 0.01 |
| | 40-50°N | p < 0.01 | | p < 0.01 | p < 0.01 | p < 0.01 |
| | 50-60°N | p < 0.01 | p < 0.01 | | p < 0.01 | p < 0.01 |
| | 60-70°N | p < 0.01 | p < 0.01 | p < 0.01 | | p < 0.01 |
| | 70-80°N | p < 0.01 | p < 0.01 | p < 0.01 | p < 0.01 | |


**Table A3.** Significance values for zonal linear trends derived from a Monte-Carlo test comparison for
mean annual temperatures ($T_{ann}$).

| | | 30-40°N | 40-50°N | 50-60°N | 60-70°N | 70-80°N |
|---|---|---|---|---|---|---|
| **Western North America** | 30-40°N | | p < 0.01 | p < 0.01 | p < 0.01 | p < 0.01 |
| | 40-50°N | p < 0.01 | | p < 0.01 | p < 0.01 | p < 0.01 |
| | 50-60°N | p < 0.01 | p < 0.01 | | p < 0.01 | p < 0.01 |
| | 60-70°N | p < 0.01 | p < 0.01 | p < 0.01 | | p < 0.01 |
| | 70-80°N | p < 0.01 | p < 0.01 | p < 0.01 | p < 0.01 | |
| **Eastern North America** | 30-40°N | | p < 0.01 | p < 0.01 | p < 0.01 | p < 0.01 |
| | 40-50°N | p < 0.01 | | p < 0.01 | p < 0.01 | p < 0.01 |
| | 50-60°N | p < 0.01 | p < 0.01 | | p < 0.01 | p < 0.01 |
| | 60-70°N | p < 0.01 | p < 0.01 | p < 0.01 | | p < 0.01 |
| | 70-80°N | p < 0.01 | p < 0.01 | p < 0.01 | p < 0.01 | |
| **Europe** | 30-40°N | | p < 0.01 | p < 0.01 | p < 0.01 | p < 0.01 |
| | 40-50°N | p < 0.01 | | p < 0.01 | p < 0.01 | p < 0.01 |
| | 50-60°N | p < 0.01 | p < 0.01 | | p < 0.01 | p < 0.01 |
| | 60-70°N | p < 0.01 | p < 0.01 | p < 0.01 | | p < 0.01 |
| | 70-80°N | p < 0.01 | p < 0.01 | p < 0.01 | p < 0.01 | |
| **Asia** | 30-40°N | | p < 0.01 | p < 0.01 | p < 0.01 | p < 0.01 |
| | 40-50°N | p < 0.01 | | p < 0.01 | p < 0.01 | p < 0.01 |
| | 50-60°N | p < 0.01 | p < 0.01 | | p < 0.01 | p < 0.01 |
| | 60-70°N | p < 0.01 | p < 0.01 | p < 0.01 | | p < 0.01 |
| | 70-80°N | p < 0.01 | p < 0.01 | p < 0.01 | p < 0.01 | |


**Table A4.** Significance values for zonal linear trends derived from a Monte-Carlo test comparison for
annual precipitation ($P_{ann}$).

|  |  | 30-40°N | 40-50°N | 50-60°N | 60-70°N | 70-80°N |
|---|---|---|---|---|---|---|
| **Western North America** | 30-40°N |  | p < 0.01 | p < 0.01 | p < 0.01 | p < 0.01 |
|  | 40-50°N | p < 0.01 |  | p < 0.01 | p < 0.01 | p < 0.01 |
|  | 50-60°N | p < 0.01 | p < 0.01 |  | p < 0.01 | p < 0.01 |
|  | 60-70°N | p < 0.01 | p < 0.01 | p < 0.01 |  | p < 0.01 |
|  | 70-80°N | 0.06 | p < 0.01 | p < 0.01 | p < 0.01 |  |
| **Eastern North America** | 30-40°N |  | p < 0.01 | p < 0.01 | p < 0.01 | p < 0.01 |
|  | 40-50°N | p < 0.01 |  | p < 0.01 | p < 0.01 | p < 0.01 |
|  | 50-60°N | p < 0.01 | p < 0.01 |  | p < 0.01 | p < 0.01 |
|  | 60-70°N | p < 0.01 | p < 0.01 | p < 0.01 |  | p < 0.01 |
|  | 70-80°N | p < 0.01 | p < 0.01 | p < 0.01 | p < 0.01 |  |
| **Europe** | 30-40°N |  | p < 0.01 | p < 0.01 | p < 0.01 | p < 0.01 |
|  | 40-50°N | p < 0.01 |  | p < 0.01 | p < 0.01 | p < 0.01 |
|  | 50-60°N | p < 0.01 | p < 0.01 |  | p < 0.01 | p < 0.01 |
|  | 60-70°N | p < 0.01 | p < 0.01 | p < 0.01 |  | p < 0.01 |
|  | 70-80°N | p < 0.01 | p < 0.01 | p < 0.01 | p < 0.01 |  |
| **Asia** | 30-40°N |  | 0.08 | p < 0.01 | p < 0.01 | 0.76 |
|  | 40-50°N | 0.02 |  | p < 0.01 | p < 0.01 | p < 0.01 |
|  | 50-60°N | p < 0.01 | p < 0.01 |  | p < 0.01 | p < 0.01 |
|  | 60-70°N | p < 0.01 | p < 0.01 | p < 0.01 |  | p < 0.01 |
|  | 70-80°N | 0.39 | 0.02 | p < 0.01 | p < 0.01 |  |


**Table A5.** Significance values for continental means linear trends derived from a Monte-Carlo test
comparison.

| | | Western North America | Eastern North America | Europe | Asia |
|---|---|---|---|---|---|
| $T_{July}$ | Western North America | | $p < 0.01$ | $p < 0.01$ | $p < 0.01$ |
| | Eastern North America | $p < 0.01$ | | $p < 0.01$ | $p < 0.01$ |
| | Europe | $p < 0.01$ | $p < 0.01$ | | $p < 0.01$ |
| | Asia | $p < 0.01$ | $p < 0.01$ | $p < 0.01$ | |
| $T_{ann}$ | Western North America | | $p < 0.01$ | $p < 0.01$ | $p < 0.01$ |
| | Eastern North America | $p < 0.01$ | | $p < 0.01$ | $p < 0.01$ |
| | Europe | $p < 0.01$ | $p < 0.01$ | | 0.08 |
| | Asia | $p < 0.01$ | $p < 0.01$ | 0.9 | |
| $P_{ann}$ | Western North America | | $p < 0.01$ | $p < 0.01$ | $p < 0.01$ |
| | Eastern North America | $p < 0.01$ | | $p < 0.01$ | $p < 0.01$ |
| | Europe | $p < 0.01$ | $p < 0.01$ | | $p < 0.01$ |
| | Asia | $p < 0.01$ | $p < 0.01$ | $p < 0.01$ | |


**Table A6.** Significance values for continental means compared to the Northern Hemispheric mean
derived from a Monte-Carlo test comparison.

| | Western North America | Eastern North America | Europe | Asia |
|---|---|---|---|---|
| $T_{July}$ | $p < 0.01$ | $p < 0.01$ | $p < 0.01$ | $p < 0.01$ |
| $T_{ann}$ | $p < 0.01$ | $p < 0.01$ | $p < 0.01$ | $p < 0.01$ |
| $P_{ann}$ | $p < 0.01$ | $p < 0.01$ | $p < 0.01$ | $p < 0.01$ |

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
