# Peer review of "Regional pollen-based Holocene temperature and precipitation patterns depart from the Northern Hemisphere mean trends"

_EGUsphere, 2022_

## Author Comment (AC1)

**Regional pollen-based Holocene temperature and precipitation patterns depart from the Northern Hemisphere mean trends**

**Response to comments of Anonymous Referee #1**

**1. General comments**

**Reviewer comment: (1)** *Herzschuh and colleagues present a very nice set of Holocene pollen-based reconstructions of $T_{ann}$, $T_{July}$, and $P_{ann}$ from 1676 sites from the Northern Hemisphere extra-tropics in order to characterize the continental, latitudinal, and regional patterns of Holocene temperature and precipitation changes in the Northern Hemisphere extra-tropics. This synthesis study is excellent and allows for the regional heterogeneity of the temperature and precipitation trend to be mapped.*
**Response:** Thank you for this nice comment.

**2. Major issues**
**2.1 quality and accuracy of the datasets**

**Reviewer comment: (2)** *The selection of records in the dataset for the Holocene quantitative reconstruction in this paper is unclear. The quality and accuracy of the synthesis studies depends largely on the chronological framework, archive type, sampling resolution of the original fossil pollen records, and so on. I note that those 991 records do cover the full period of 11 to 1 ka, but do not see an evaluation of the age, resolution, and archive type of the selected records. Are there any selection criteria for chronology and archive type in the dataset? For example, how many age control points does the original record contain that will be selected for quantitative reconstruction? And what is the time resolution of each sample? This takes into account that the amplitude of changes in temperature and precipitation reconstructions would vary substantially with the resolution of the proxy record.*
**Response:** This is described in Herzschuh et al. (2022b). We restructured and added additional explanations from the selection process, i.e. about archive types and chronologies to the text in the methods section.
**New text:** This study analyzes pollen-based reconstructions provided in the LegacyClimate 1.0 dataset (Herzschuh et al., 2021). It contains pollen-based reconstructions of $T_{July}$, $T_{ann}$, and $P_{ann}$ of 2593 records along with transfer function metadata and estimates of reconstruction errors and is accompanied by a manuscript analyzing reconstruction biases and presenting reliability tests (Herzschuh et al., 2022a). The fossil pollen records, representing the LegacyPollen 1.0 dataset, were derived from multiple natural archives, most commonly continuous lacustrine and peat accumulations (Herzschuh et al., 2022b), and originate from the Neotoma Paleoecology Database (´Neotoma´ hereafter; last access: April 2021; Williams et al., 2018), a dataset from Eastern and Central Asia (Cao et al., 2013; Herzschuh et al., 2019), a dataset from Northern Asia (Cao et al., 2019), and a few additional records to fill up some spatial data gaps in Siberia.

The chronologies of LegacyPollen 1.0 are based on revised ´Bacon´ (Blaauw and Christen, 2011) age-depth models with calibrated ages at each depth provided by Li et al. (2022). Taxa are harmonized to genus level for woody and major herbaceous taxa and to family level for other herbaceous taxa. Along with LegacyClimate 1.0, a taxonomically harmonized modern pollen dataset is provided (a total of

15379 samples; Herzschuh et al., 2022a) which includes datasets from Europe (EMPD2, Davis et al., 2020), Asia (Tarasov et al., 2011; Herzschuh et al., 2019; Dugerdil et al., 2021), and North America (from Neotoma; Whitmore et al., 2005). LegacyClimate 1.0 also provides the climate data for the sites of the modern pollen samples that were derived from WorldClim 2 (Fick and Hijmans, 2017).

[...]

Of the 2593 records available in LegacyClimate 1.0, 1908 records with at least 5 samples that cover at least 4000 years of the Holocene and have a mean temporal resolution of 1000 years or less were included in the time-slice comparisons based on this criterion (Fig. 1). The construction of time-series to estimate the means of climate variables was further restricted to 957 records that cover the full period of 11 to 1 ka.

**Reviewer comment: (3)** *In addition, the range and quantity of selected modern sites in the calibration dataset can also affect the accuracy of temperature and precipitation reconstructions, as suggested by the authors. Then how many transfer functions are used to calculate the 991 records in this synthesis study? Does each record need to establish a transfer function, or does it establish by region? Is the spatial range of modern sites in the calibration dataset for establishing transfer function all within a 2000 km radius? Or are there some differences in different continents or regions? Of course, I believe that the trend of paleo-temperature and paleo-precipitation change will not change substantially, but it will affect the comparison of amplitude.*

**Response:** For each record, its own calibration data set was established by including all modern samples from within a 2000 km radius around the side. This is described in the text. More details are provided in the ESSD manuscript about the LegacyClimate 1.0 dataset (Herzschuh et al., 2022a; in discussion).

**New text:** For each fossil site, we calculated the geographic distance between each modern sampling site and each fossil location and selected a unique calibration set from modern sites within a 2000 km radius (Cao et al., 2014), as it was shown to be a good trade-off between analog quality and quantity (Cao et al., 2017).

**2.2 effect of correlation between temperature and precipitation reconstructions**

**Reviewer comment: (4)** *I agree with the authors that "Pollen data are one of the few land-derived proxies available that can theoretically contain independent information on both temperature and precipitation in the same record" (Lines 99-101). Therefore, the authors reconstructed the spatio-temporal patterns of temperature and precipitation from a single dataset simultaneously. However, it is a challenge to distinguish the effect and correlation between temperature and precipitation in quantitative analysis. In the section of Methods and Discussions, the author mentions the issue of the impact of precipitation on temperature reconstruction (Lines 143-145, 410-412). Could you give more explanation as to why such an approach would 'restrict the impact of precipitation on temperature reconstruction and vice versa'? One or two sentences will do.*

**Response:** We added some explanation in the methods section.

**New text:** A WA-PLS_tailored reconstruction is also provided in the LegacyClimate 1.0 dataset (Herzschuh et al., 2022a), which addresses the problem that co-variation in modern temperature and precipitation data can be transferred into the reconstruction. To reduce the influence of one climate variable to the target variable, the modern range of the non-target variable is reduced by tailoring the modern pollen dataset to a selection of sites with little covariance between the two variables. For

example, to reconstruct $T_{July}$ we identified the $P_{ann}$ range reconstructed by WA-PLS and extended it by 25% at both ends. For the selection of sites in the modern training dataset, we then restricted modern $P_{ann}$ to that range accordingly. As such, we keep all information for reconstruction from those modern pollen spectra that cover a wide temperature range but downweight the information from pollen spectra covering a wide precipitation range.

**Reviewer comment: (5)** *In addition, how do the effects of temperature and precipitation on each other differ across continents and regions? How is it evaluated in quantitative reconstruction analysis?*

**Response:** The effects of temperature and precipitation on each other are evaluated for each site separately, e.g. with a statistical significance test (Telford and Birks, 2011) by partialling out the explained variance in the pollen data by the respective other variables. We made assessments of the significance tests in Herzschuh et al. (2022a). However, the effectiveness of obtaining independent reconstructions depends on how strongly correlated temperature and precipitation are in the modern training data set.

**2.3 reconstruction uncertainty**

**Reviewer comment: (6)** *I do not see the expression of reconstruction uncertainty in Figures 2 and 5. The evaluation of the reconstruction errors is essential for quantitative reconstruction and comparisons of different results. Therefore, it would be appropriate to add each latitudinal reconstruction curve with 1σ uncertainty shaded to the supplementary file.*

**Response:** The focus of the main text Figures 3 and 6 is on the (red) mean curves, for which we added shading corresponding to the standard error, i.e. $\sigma/\sqrt{n}$, on the mean of the records; the standard error is calculated taking into account the same weighting scheme as applied to calculate the mean. We follow your suggestion and provide the uncertainty ranges of all latitudinal means from records assessed in this study for the Northern Hemisphere and all (sub-)continents in the Appendix (Appendix Figures 3-7).

**Reviewer comment: (7)** *The number of records (n) for each curve in Figures 2 and 5 also needs to be displayed in the appropriate place. Is the large range of temperature and precipitation variations in North America north of 60°N caused by the number of records (n)?*

**Response:** Latitudinal curves are only shown and included in the calculation of the weighted mean curve in Figures 3 and 6, if they include more than 3 grid cells in order to reduce the influence of latitudinal bands with very few records. Labels in corresponding colors to the latitudinal curves were added to each panel in the figures to indicate the number of grid boxes that contributed to each latitudinal curve, so that the readers can bring them in relation to the variation in the curves.

**2.4 Holocene temperature conundrum**

**Reviewer comment: (8)** *There is still a great controversy regarding the occurrence of Holocene thermal maximum between the proxy temperature reconstructions and climate models, named as "Holocene temperature conundrum". One of the main controversies for Holocene temperature conundrum is the occurrence of a maximum in mean annual temperature (MAT) during the early to middle Holocene. The term 'mid-Holocene optimum/late-Holocene optimum' has been used in this paper, but in some areas, such as East Asia, there is a difference between mid-Holocene optimum and Holocene warm*

*period/Holocene thermal maximum. Mid-Holocene optimum is thought to be a period of high temperature and high precipitation, when vegetation flourishes. The authors should define the mid-Holocene optimum and distinguish it from the Holocene warm period.*

**Response:** Thank you for your suggestion regarding the wording of Mid-/Late Holocene optimum. We changed the wording to "Holocene temperature maximum" and "Holocene precipitation maximum" to address the timing of the maximum values in the reconstruction and distinguish them from the Holocene warm period.

**Reviewer comment: (9)** *In addition, quantitative Holocene temperature records in East Asia (loess, lakes, marine sediments) reveal a clear early to middle Holocene thermal maximum, such as high-resolution Holocene pollen records from Xiaolongwan Maar Lake in northeastern China, Gonghai Lake in northern China, and Huguangyan Maar Lake in southern China. These records show the occurrence of a maximum in MAT during the early to middle Holocene, which does not support the conclusion of this paper that "The concept of a mid-Holocene temperature optimum only applies mainly to the mid and high northern latitudes in the circum-North Atlantic region while records from mid-latitude Asia, Western North America, and all subtropical areas do not fit into this concept but mostly show an overall Holocene increase or other pattern" (Lines 430-434).*

**Response:** Cao et al. (2017) discuss the influence of the modern pollen-climate calibration dataset extent on climate reconstructions on three pollen records from China. They conclude that a careful selection of the spatial extent of the calibration dataset has a significant influence on the reconstruction performance, i.e. that small-scale calibration datasets might not include enough spatial variation in modern pollen assemblages to cover the temporal variation of the target fossil pollen record. The selected spatial extent led to a temporal pattern in mid-latitude Asia presented in this study.

- Cao, X., Tian, F., Telford, R. J., Ni, J., Xu, Q., Chen, F., Liu, X., Stebich, M., Zhao, Y., Herzschuh, U.,: Impacts of the spatial extent of pollen-climate calibration-set on the absolute values, range and trends of reconstructed Holocene precipitation. Quaternary Science Reviews 178, 37-53. https://doi.org/10.1016/j.quascirev.2017.10.030, 2017.

**New text (methods section):** For each fossil site, we calculated the geographic distance between each modern sampling site and each fossil location and selected a unique calibration set from modern sites within a 2000 km radius (Cao et al., 2014), as it was shown to be a good trade-off between analog quality and quantity (Cao et al., 2017).

**New text (discussion section):** Our pollen-based reconstructions are all performed with WA-PLS, which is known to produce smaller climate amplitudes than MAT (a likewise commonly used method) because it is less sensitive to extreme climate values in the modern pollen dataset (Birks and Simpson 2013; Cao et al., 2017; Nolan et al. 2019). Furthermore, by using a standard area size for our modern pollen datasets, we may have stabilized the regional reconstructions, that is, equalized the amplitude as the source areas represent rather similar biogeographical and climate ranges.

**3. Minor issues**

**Reviewer comment: (10)** *L202: for "in Europe north of 60°C" consider "in Europe north of 60°N".*

**Response:** Thank you for your suggestion! This was a typo that we fixed now.

**Reviewer comment: (11)** *L275: for "~0.07K compared to ~0.18K" consider "…°C compared to…".*
**Response:** We changed the unit from K to °C.

**Reviewer comment: (12)** *L336: "…the modern pollen assemblages are not heavily biased by human impact", please provide relevant literature here.*
**Response:** the modern pollen assemblages are assumed to not be heavily biased by human impact, however, this is not true for all regions. We inferred the effect of human impact from the abundance of Plantaginaceae and *Rumex* as indicators of grazing and such intense animal husbandry and identified regions that are potentially biased by human influence in Herzschuh et al. (2022a) and further discussed this in the discussion section.
**New text:** As with all applications of taxa-based transfer functions to fossil records, we assume that both modern and past taxa assemblages (in our case, vegetation) are in equilibrium with climate, and that the relationships inferred from modern data do not change throughout the Holocene (Birks et al., 2010; Chevalier et al., 2020) and that the modern pollen assemblages are not heavily biased by human impact. Differences in global boundary conditions during the Early to Mid-Holocene (e.g., lower atmospheric $CO_2$ concentration, different seasonal insolation) however, may have modified these relationships, which could have also dampened the reconstructed amplitudes. Also, vegetation response to climate change may be involve lags (see the ongoing discussion about the so-called ´forest conundrum´, i.e., the observation that observed forest maximum lags the simulated temperature maximum; Dallmeyer et al., 2022) and depends on the initial conditions such as the distribution of refugia during the Last Glacial (Herzschuh et al., 2016; 2020). Furthermore, there are areas, especially the densely settled regions in Europe and Southeastern Asia, that are affected by human activities throughout the Holocene due to intense animal husbandry, as inferred from the abundance of Plantaginaceae and *Rumex* as indicators of grazing (Herzschuh et al., 2022a), or due to industrialization since the second half of the 19th century. This probably led to extinction events, especially for disturbance-dependent taxa and contributed to gaps within the potential bioclimatic space of taxa that form natural communities (Zanon et al., 2018).

**Reviewer comment: (13)** *L362: for "from the early to mid-Holocene" consider "from the middle to late-Holocene".*
**Response:** In fact, "from the early to mid-Holocene" is correct - from our reconstructions, we found a high precipitation in the Early and Mid-Holocene in East Asia (Fig. 4: %Pann 9 ka minus 6 ka). For sites in Central Asia, a decline in precipitation in the Early Holocene can be reported for 11 ka minus 9 ka (indicated by red colors). We admit that the original sentences were not clearly stated and we rephrased the paragraph to make it more clear and also point to the panels in Fig. 4 that we refer to.

**Reviewer comment: (14)** *Figures 3 and 4: The map would be improved by changing some colors and size. Each 2°x2° grid cell was too small to see, even zoomed in. Changing sizes of maps and/or colors may resolve this better.*
**Response:** We revised the maps in the Figures 4 and 5, made them bigger, changed the arrangement of the panels, reduced white space between the panels, removed redundant labeling and improved the color contrast (see also Reviewer comment (14) of Referee #2).

---

## Author Comment (AC2)

**Regional pollen-based Holocene temperature and precipitation patterns depart from the Northern Hemisphere mean trends**

**Response to comments of Referee #2**

**1. Major issues**

**1.1 Significance of the reconstructions**

**Reviewer comment: (3)** *Significance of the reconstructions. Tests in H2022 show that only approximately â of the reconstructions show a temperature trend that deviates from noise (i.e. where the reconstruction shows a better correlation with the first principle component of the assemblages than 90 % of the reconstruction based on randomized temperatures; Table 2 in H2022). In the absence of any information about this in the method section, I assume that the same proportion holds for the selection of time series analyzed here. So, why did the authors not filter out these records, as was for instance done in previous work (Marsicek et al. 2018)? As it stands, the analysis presented here is based on reconstructions that are for approximately 66 % noise. Thus the authors really need to convince the reader why they ignore their own previous analyses and present the evidence they have that these reconstructions are valid. One obvious way to do so would be using sensitivity tests and to assess to what degree the observed trends are sensitive to the significance of the individual time series. (If on the other hand, the authors argue that these tests are not meaningful for assessing the robustness of the reconstruction, then that needs to be reflected in H2022.)*

**Response:** Results of a significance test sensu Telford & Birks (2011) are presented in Herzschuh et al., (2022a); the significance test shows rather low percentages of records that are significant. However, it is discussed in the literature that the Telford-Birks-test is rather conservative and that several other reasons could potentially cause a reconstruction to be flagged as non-significant (see Herzschuh et al., 2022a). A visual inspection of the latitudinal means between those reconstructions derived from WA-PLS, WA-PLS_tailored and WA-PLS_tailored with significant records revealed rather similar overall patterns, which suggests that non-significant records don't affect the outcome of our analyses presented in this study. We provide plots with the latitudinal means WA-PLS_tailored and WA-PLS_tailored with significant records similar to Fig. 3 in the Appendix (Appendix Figures 1 and 2).

**1.2 co-variation of temperature and precipitation reconstructions**

**Reviewer comment: (4)** *Here and in H2022 the authors discuss the independence of the temperature and precipitation reconstructions. This is an important issue as the second aim of this study is "What are the continental, latitudinal, and regional patterns of Holocene precipitation change and how do these changes co-vary with temperature trends?" (L111-112). In H2022 the authors use a method to reduce the influence of covariance between temperature and precipitation (tailoring). They conclude that "The tailoring successfully reduced the co-variation of temperature and precipitation in the modern dataset as indicated by the distribution of the correlation coefficient in Fig. 8. Nevertheless, the obtained reconstructions are largely consistent between WA-PLS and WA-PLS-tailored: a correlation of r >= 0.9 is found for 59.2% of all records for TJuly, 60.7% for Tann and 56.5% for Pann." (L292-296 H2022). Notwithstanding whether the r >= 0.9 is a good criterion or not, my conclusion is that the tailored reconstructions are superior because they suffer less from co-variation and that about*

*40 % of the time series are markedly different from the non-tailored ones. So if independence of the temperature and precipitation reconstructions is a concern, I fail to understand why the authors ignore their own solution to this problem and not simply use the tailored reconstructions. Similarly, how independent are the annual and July temperature estimates and can one really interpret the difference between them?*

**Response:** We compared the latitudinal mean curves derived from the reconstruction with WA-PLS (Fig. 3) with those curves derived from the reconstruction with WA-PLS_tailored (Appendix Figure 1) and found similar patterns. Hence, we decided to use the standard WA-PLS-derived reconstruction to be consistent with previous studies (see methods section). We conclude that co-variation between temperature and precipitation in the modern calibration dataset is not a major issue in the reconstructions.

In Herzschuh et al. (2022a) we applied a Canonical Correlation Analysis (CCA) to the modern training dataset to infer the relationship between the modern pollen assemblages and climate. A high ratio (>= 1) of constrained ($\lambda_1$) and unconstrained ($\lambda_2$) explained variance indicate ecologically important determinants. We found the spatial pattern of $\lambda_1/\lambda_2$ for $T_{ann}$ overall similar to $T_{July}$, but with slightly higher values. We reconstructed both, $T_{ann}$ and $T_{July}$, as authors use either mean annual temperatures or seasonal (e.g. $T_{July}$) temperatures for synthesis studies and model-data comparisons. Therefore, we provide both temperature estimates so that the authors could choose which variable they want to use. For our analyses in this study, we also applied our Monte-Carlo test comparison to assess if the linear trends between the reconstructed climate variables are significantly different.

**1.3 age uncertainty of the reconstructions**

**Reviewer comment: (5)** *Furthermore, the authors mention the reconstruction uncertainty in the method section and refer to LegacyAge 1.0 (https://doi.org/10.5194/essd-2021-212) for the chronology (and its uncertainties). It remains nevertheless unclear how these uncertainties are treated or if they are considered at all in the analyses presented here. This is important as the inferred changes in temperature and precipitation are small relative to the stated error and because LegacyAge 1.0 indicates that age uncertainties of the time series have a median uncertainty of about 500 years (but reach to over 1,000 years). So, are the regional reconstructions really different from each other?*

**Response:** Using the full reconstruction error would be over-conservative as the errors are not independent and a large part of the stated error will be in the form of a constant bias for all the samples in a given record, which will then vanish when taking the anomalies. It remains an unresolved issue in the field to our knowledge. Regarding the chronological errors, as they are independent between sites, their overall contribution to a regional average will be small. The same will be true for the reconstruction errors for large enough regions. We now show the regional reconstructions with the standard error computed from the spread between the records. In addition, we applied a Monte-Carlo test comparison to examine linear trends of the latitudinal means and test if they are significantly different from each other (see methods section). We tested the linear trends for both, the zonal means within the continents as well as the weighted means between the continents (see Appendix Tables 2-5).

**1.4 reconstruction errors**

**Reviewer comment: (6)** *L156-160: "As it has already been shown in previous comparisons, WA-PLS can have higher RMSEPs than MAT but these do not necessarily reflect a less reliable reconstruction but methodological differences (Cao et al., 2014)." This is an interesting statement and it would be good to repeat some of the reasoning presented in Cao et al here. More importantly, if the estimate of the error is method dependent, how useful then is the error? Would one not get a better, more meaningful, estimate of the reconstruction uncertainty if the difference between various methods is accounted for (see e.g. (Kaufman, McKay, Routson, Erb, Dätwyler, et al. 2020)).*

**Response:** We added a sentence to the text with some of the reasoning presented in Cao et al. (2014). As for the estimation of the reconstruction uncertainties, we presented quantitative pollen-based reconstructions with 3 different methods (i.e., WA-PLS, WA-PLS_tailored and MAT) in Herzschuh et al. (2022a) and made assessments about estimating the reconstruction uncertainty. However, for this study, we focus on WA-PLS and WA-PLS_tailored, so it would make little sense to derive the reconstruction errors from a comparison of those two methods.

**New text:** As it has already been shown in previous comparisons, WA-PLS can have higher RMSEPs than MAT but these do not necessarily reflect a less reliable reconstruction but methodological differences. MAT is known to be more sensitive to spatial autocorrelation, which causes the model performance to be over-optimistic compared to WA-PLS (Cao et al., 2014).

**Reviewer comment: (7)** L160-161: "*Besides, the reconstruction errors are likely much smaller when only the trends and the relative changes are assessed, as in this study.*" This may be true to some extent, but it would be good if the authors provided some explanation for this statement.

**Response:** Arguments are mainly that the same transfer function (based on the same modern dataset) is used and errors are not independent from each other. However, a quantification of the dependence remains, to our knowledge, an unsolved problem. We expand the explanation.

**New text:** Besides, trends and the relative changes, as interpreted in this study, are less sensitive to methodological biases than absolute values.

**1.5 methodology**

**Reviewer comment: (8)** *Finally, the section on methodology to calculate the time series of temperature and precipitation is descriptive, but I have some additional questions and, crucially, miss some explanation of the rationale. Why were the time series 500-year smoothed and resampled at 100 year resolution and spatially averaged (at 2x2 deg) prior to analysis and why is that the best method if one aims to investigate spatial variability? How was the value of 500 years chosen? How close is the 100 years to the actual resolution of the time series? How were gaps in the time series treated (looking at the data at https://doi.pangaea.de/10.1594/PANGAEA.930500 it appears that the sampling was not done continuously in depth and the time series therefore contain gaps). How spatially representative are the averaged time series for the different subsets? I.e. how many time series (or 2x2 grid cells) used for each regional reconstruction? And how does data availability affect the (un)certainty of the reconstructions and the differences among them?*

**Response:** The focus of this study is to assess temporal variability on a multi-decadal to centennial-scale and therefore we are interested in long-term trends. To infer those long-term trends, we applied a 500-year smoothing to the time series, which is the typical resolution of the time series used in this study. The function that we used for the smoothing is designed to resample irregularly sampled time

series to an equidistant spacing (Reschke et al., 2019) so that we can read climate values for each record at the exact same time-slice. To address how many time series contribute to a single grid cell, we added a map to the manuscript (Fig. 2).

**2. Minor issues**

**Reviewer comment: (9)** *L79-80: "despite the existence of many Holocene pollen records" this seems an odd comment given that some of the syntheses referred to in this sentence post-date the "previous reconstructions". Moreover, some of the authors of this study were also involved in the temperature 12k project, raising the question why they did not include these records at that time.*

**Response:** The phrasing of this sentence was misleading, indeed. We rephrased the sentence.

**New text:** Synthesis studies hitherto included rather few records from the large non-glaciated Asian continent (Andreev et al., 2004; Leipe et al., 2015; Melles et al., 2012; Nakagawa et al., 2002; Stebich et al., 2015; Tarasov et al., 2009 and 2013). The inclusion of recently compiled Holocene pollen records (Cao et al., 2019; Herzschuh et al., 2019) and high-quality modern pollen datasets (Tarasov et al., 2011; Cao et al., 2014; Davis et al., 2020; Dugerdil et al., 2021) from Asia now allows for higher quality quantitative reconstructions.

**Reviewer comment: (10)** *L154: rather than referring to a map that shows the error for the entire LegacyClimate dataset, it would be helpful to present a map of the reconstruction error for the subset of ~1600-900 time series analyzed here.*

**Response:** We think it is sufficient to refer to the error maps in Herzschuh et al. (2022a) as the 957 records presented here are only a subset of the LegacyClimate 1.0 dataset and the spatial pattern would not change if we present such maps with only the subset. Therefore, we don't see the necessity to add an additional figure (given that our main text and Appendix is already so figure-heavy).

**Reviewer comment: (11)** *L176: please define the boundary between Asia and Europe.*

**Response:** We defined the boundary between Asia and Europe at 43°E between Black Sea and Caspian Sea. We indicated the boundaries between the continents in the text now.

**New text:** To calculate zonal, (sub-)continental (i.e., Asia (>43°E), Europe (<43°E), Eastern North America (<104°W; Williams et al., 2000) and Western North America), and hemispheric means we selected all 957 smoothed and resampled time-series of $T_{July}$, $T_{ann}$, and $P_{ann}$ that cover the full period between 11 and 1 ka and calculated climate anomalies for all three climate variables.

**Reviewer comment: (12)** *L247: "fewer" instead of "less".*

**Response:** Thank you, we changed the wording.

**Reviewer comment: (13)** *L250: "values outside the range" please show (or mention) the entire range and what proportion of the data points falls within the restricted range. This sentence raises suspicion about the reconstructions that can easily be avoided.*

**Response:** We added a table with the entire ranges and the proportions of values that fall within the restricted range in the Appendix (Appendix Table 1), as requested.

**Reviewer comment: (14)** *The maps, especially those in Fig. 3, are really small and difficult to read. The individual panels can be made bigger by removing white space and redundant labeling without the need to increase the overall figure size.*

**Response:** We revised the maps in Figures 1, 4 and 5, made them bigger, changed the arrangement of the panels, reduced white space between the panels, removed redundant labeling and improved the color contrast (see also Reviewer comment (14) of Referee #1).